# Democratic actions with scalar fields:
# Symmetric sigma models, supergravity actions and
# the effective theory of the type IIB superstring

Jose Juan Fernandez-Melgarejo[1][*], Giacomo Giorgi[1][†], Carmen Gomez-Fayren[2][‡],
Tomas Ortin[2][°] and Matteo Zatti[2][§]

**1** Departamento de Electromagnetismo y Electrónica, Universidad de Murcia,
Campus de Espinardo, 30100 Murcia, Spain
**2** Instituto de Física Teórica UAM/CSIC, C/ Nicolás Cabrera, 13–15,
C.U. Cantoblanco, E-28049 Madrid, Spain

[*] melgarejo@um.es , [†] giacomo.giorgi@um.es , [‡] carmen.gomez-fayren@estudiante.uam.es ,
[°] tomas.ortin@csic.es , [§] matteo.zatti@estudiante.uam.es

## Abstract

The dualization of the scalar fields of a theory into $(d-2)$-form potentials preserving all the global symmetries is one of the main problems in the construction of democratic pseudoactions containing simultaneously all the original fields and their duals. We study this problem starting with the simplest cases and we show how it can be solved for scalars parametrizing Riemannian symmetric $\sigma$-models as in maximal and half-maximal supergravities. Then, we use this result to write democratic pseudoactions for theories in which the scalars are non-minimally coupled to $(p+1)$-form potentials in any dimension. These results include a proposal of democratic pseudoaction for the generic bosonic sector of 4-dimensional maximal and half-maximal ungauged supergravities. Furthermore, we propose a democratic pseudoaction for the bosonic sector of $\mathcal{N} = 2B, d = 10$ supergravity (the effective action of the type IIB superstring theory) containing two 0-, two 2-, one 4-, two 6- and three 8-forms which is manifestly invariant under global $SL(2, \mathbb{R})$ transformations.

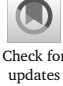

# 1  Introduction

$p$-branes naturally (*electrically*) couple to $(p + 1)$-form potentials [1–12].[1] However, the theories that describe the bulk dynamics of those $(p + 1)$-form potentials (supergravity theories, typically) are usually written in terms of the lowest-rank potentials which are dual to them. Thus, $\mathcal{N} = 2A, d = 10$ supergravity, the effective fields theory of the type IIA superstring, is usually written in terms of the metric, dilaton (a scalar), NSNS 2-form, and RR 1- and 3-forms plus a mass parameter (Romans') while the solitonic 5-brane couples to the NSNS 6-form dual to the 2-form, and the D4-, D6- and D8-branes couple to RR 5-, 7- and 9-forms dual, respectively, to the RR 3- and 1-forms and to the mass parameter.

Defining the higher-rank forms needed to describe the couplings of higher-dimensional branes is always possible on-shell, providing equations of motion for all of them. It is always desirable to have an action from which those equations of motion can be derived.[2] However, the field strengths of the higher-rank forms typically contain the lower-rank ones and, very often, it is not possible to find an action for the higher-rank forms because it must contain, at the same time, the lower-rank ones, which are related to the former in a highly non-local way through the duality relations.

The fact that an action for the higher-rank forms must also contain the lower-rank ones can be turned into an advantage if one manages to give consistence to the simultaneous presence of dual fields in the action.[3] A solution to this problem is to use extensions of the Pasti-Sorokin-Tonin formalism [22] which is based on the introduction of an auxiliary scalar field in the action. This method has been used in Ref. [23, 24][4] to construct covariant actions of $\mathcal{N} = 2B, d = 10$ supergravity containing the 4-form but also 8-form duals of the scalar fields and also in Refs. [27, 28] to construct an action of $\mathcal{D} = 1, d = 11$ supergravity containing the 3-form and its dual 6-form simultaneously. By dimensional reduction one can obtain an action of $\mathcal{N} = 2B, d = 10$ supergravity containing the fundamental and dual fields [29]. In a slightly different context, it has been used in Ref. [30] to construct a covariant worldvolume action of the M5-brane.

---

[1]More references can be found in the reviews [13, 14].

[2]There is another reason why one may need the presence of the higher-rank forms in the action: in flux compactifications, their fluxes make relevant contributions [15]. In particular, in Refs. [16, 17] it is manifestly shown that, under the presence of D$p$/O$p$ systems, the modifications of the electric field strengths (and their Bianchi identities) induced by open string fluxes are read off from the couplings of the dual potentials to such objects.

[3]See Ref. [18] for the case of nonlinear electrodynamics and Ref. [19] for the extension to $(p+1)$-form potentials and their duals in arbitrary dimensions. The results of Ref. [19] could be used to formulate proper actions for the systems described in the current work via pseudoactions. For the free fields, this approach was introduced in Ref. [20] and discussed in detail for arbitrary $(p + 1)$-form potentials and their duals in arbitrary dimensions in Ref. [21].

[4]See Ref. [25] and, specially, the more recent Ref. [26] for a review.

An alternative solution, proposed in Ref. [31], consists in including all the fields and treating them all on an equal footing as independent (making it *democratic*). This procedure doubles the degrees of freedom and one has to impose by hand the twisted duality [32,33] relations between $(p+1)$- and $(d-p-3)$-forms only after the equations of motion have been derived from the action. Since the duality relations are not derived from the action, one is actually dealing with a *pseudoaction*. The pseudoaction introduced in Ref. [34] for $\mathcal{N} = 2B, d = 10$ supergravity, which includes the 4-form with selfdual 5-form field strength provides a good example: it contains an unconstrained (not selfdual) 4-form which describes twice the degrees of freedom of the selfdual one and the selfduality constraint must be imposed after the equations of motion have been derived.

Each of these solutions presents advantages and disadvantages: the PST method introduces unwanted auxiliary variables but gives a proper action from which all the equations of motion can be derived while the second method does not introduce unwanted auxiliary fields but only gives a pseudoaction.[5] If one is interested in evaluating the action on-shell (in order to study black-hole thermodynamics, say), it is not clear whether the PST action gives the same value as the original one. However, the democratic pseudoaction does, in Euclidean signature, as we are going to discuss.

In this paper we are going to use the second method of dealing simultaneously with fundamental and dual fields. Thus, our goal will be to construct democratic pseudoactions containing all the fields and their duals whose equations of motion give back the original ones upon use of duality constraints. Our main concern will be the dualization of the scalar fields, which usually couple non-linearly among themselves and to other fields, into $(d-2)$-form potentials.

The standard dualization procedure is only possible when the equation of motion can be written, on-shell, as a total derivative. This happens to the equation of motion of a given scalar field when there is a global symmetry of the action acting on it, typically as a constant shift. The equation of motion, then, is equivalent to the conservation of the associated Noether current. Even if this is not immediately apparent, in that case the action can be rewritten in terms of derivatives of the scalar and one can use the Poincaré dualization method in the action. In absence of this kind of symmetry, it is not known how to dualize the scalar field, but in supergravity theories, there are typically many of these symmetries associated to dualities.

When the theory contains several scalar fields which parametrize a non-linear $\sigma$-model, things become more complicated. The shift symmetries are isometries of the $\sigma$-model metric. One can always use coordinates (scalar fields) adapted to a given isometry. In those coordinates the scalar field shifted by the isometry does not occur explicitly in the metric and the action can always be written in terms of its derivatives only. The equation of motion will be a total derivative. One can only use coordinates adapted to several isometries for those isometries that generate an Abelian subgroup. However, even if the isometries do not commute, there are many conserved currents as isometries and this guarantees that there are as many combinations of the equations of motion as isometries that can be written as total derivatives. These combinations can be used to define on-shell duals of scalars. Carrying it out this program in the full theory can be, in practice, quite complicated. See, for example, Refs. [36–38].

Often (in all $d = 4$ maximal and half-maximal supergravities, for instance), the target space is a G/H Riemannian symmetric manifold, with more isometries and conserved Noether currents than scalar fields. One can define a dual $(d-2)$-form potential associated to each of the Noether currents (see, for instance Ref. [24]), but, then, the number of dual fields, $\dim$ G, would be larger than the number of original scalar fields, $\dim$ G$-\dim$ H, which is not acceptable. However, since the set of all dual $(d-2)$-forms transform in the adjoint representation of G, removing from the action any number of them would break the global G-invariance of the theory. This is one of the problems of the democratic pseudoaction of $\mathcal{N} = 2B, d = 10$

---

[5]A more recent and different proposal can be found in Ref. [35] for $\mathcal{N} = 2A, B, d = 10$ supergravities.

supergravity proposed in Ref. [31]: only the RR scalar $C^{(0)}$ was dualized into the RR 8-form $C^{(8)}$ and, therefore, the pseudoaction is not $SL(2,\mathbb{R})$-invariant as the original theory.

A possible way out is to use a singular, but $G$-covariant, kinetic matrix in the pseudoaction, as suggested in Ref. [39]. In this paper we will identify the additional terms which are necessary to construct the complete pseudoaction and we will use this result to construct duality-invariant pseudoactions for several interesting theory, including all the $d=4$ maximal and half-maximal supergravities and $\mathcal{N}=2B, d=10$ supergravity.

We are going to consider cases of increasing complexity: in Section 2 we start with the dualization of a single, massless, real scalar $\phi$ coupled to gravity in $d$ spacetime dimensions, to establish the notation and the basic facts. In Section 3 we consider a generic non-linear $\sigma$-models with isometries and we will study the dualization of scalars associated to an Abelian subgroup. This will show us which are the needed additional terms mentioned above, which are the first interesting results of this paper. In Section 4, we study the dualization of a Riemannian symmetric $\sigma$-model and construct, using the additional terms mentioned above and the singular but G-covariant kinetic matrix suggested in Ref. [39], the democratic pseudoaction that contains the scalars that parametrize the G/H coset space and the dual $(d-2)$-form potentials while preserving the global G invariance. In Section 5 we apply this result to the case in which the scalars are coupled to $(p+1)$-form potentials, including the particular case $d=2(p+2)$, in which some of the transformations in G are electric-magnetic dualities which leave invariant the equations of motion but not the action. This particular case covers the bosonic sector of all the maximal and half-maximal 4-dimensional supergravities. Finally, in Section 6 we consider the case of $\mathcal{N}=2B, d=10$ supergravity, the effective field theory of the type IIB superstring and propose a pseudoaction that contains the dilaton and RR 0-form and a triplet of 8-forms dual to them, the $SL(2,\mathbb{R})$ doublet of 2-forms (NSNS and RR) and the dual doublet of 6-forms and a 4-form which is a $SL(2,\mathbb{R})$ singlet. The equations of all these fields derived from the pseudoaction reduce to those of the fundamental fields when the (self-) duality constraints are imposed on them.

Our conclusions and future directions of research are contained in Section 7.

## 2 Dualization of a single real scalar

In order to establish the notation and describe what we want to do, it is convenient to start with the simplest case, namely that of a single, massless, real scalar, $\phi$, coupled to gravity, described by the Vielbein $e^a = e^a{}_\mu dx^\mu$, in $d$ spacetime dimensions. The action that dictates the dynamics of this system is

$$S[e^a, \phi] = \int \left\{ (-1)^{d-1} \star (e^a \wedge e^b) \wedge R_{ab} + \tfrac{1}{2} d\phi \wedge \star d\phi \right\}.^{6} \tag{1}$$

In this action $\star$ denotes the Hodge dual and, therefore,

$$\star(e^a \wedge e^b) = \frac{1}{(d-2)!} \epsilon_{c_1 \cdots c_{d-2}}{}^{ab} e^{c_1} \wedge \cdots \wedge e^{c_{d-2}}. \tag{2}$$

$\omega^{ab} = \omega_\mu{}^{ab} dx^\mu$ is the torsionless, metric-compatible, Levi-Civita spin connection and $R^{ab} = \tfrac{1}{2} R_{\mu\nu}{}^{ab} dx^\mu \wedge dx^\nu$ is its curvature 2-form

$$R^{ab} \equiv d\omega^{ab} - \omega^a{}_c \wedge \omega^{cb}.^{7} \tag{3}$$

---

[6]In this paper we are using differential-form language and the conventions of Ref. [14].
[7]It is antisymmetric $\omega^{ab} = -\omega^{ba}$ and satisfies $De^a = de^a - \omega^a{}_b \wedge e^b = 0$.

The equations of motion which follow from this action are

$$\mathbf{E}_a = \iota_a \star (e^c \wedge e^d) \wedge R_{cd} + \frac{(-1)^d}{2} \left( \iota_a d\phi \wedge \star d\phi + d\phi \wedge \iota_a \star d\phi \right), \tag{4a}$$

$$\mathbf{E} = -d \star d\phi. \tag{4b}$$

Locally, the equation of motion of the scalar $\phi$ can be solved by introducing a $(d-2)$-form $C$ such that

$$G \equiv dC = \star d\phi. \tag{5}$$

The equation of motion of the scalar $\phi$ becomes the Bianchi identity of $G$ ($dG = 0$) and the Bianchi identity of the scalar field strength $d\phi$ ($d^2\phi = 0$) becomes the equation of motion of the dual $(d-2)$-form $C$ ($d \star G = 0$).

Observe that the field strength $G$ is invariant under gauge transformations

$$\delta_\Sigma C = d\Sigma, \tag{6}$$

where $\Sigma$ is an arbitrary $(d-3)$-form.

It is not difficult in this case to replace in the Einstein equations $d\phi$ by $\star G$ obtaining the equations of motion of a theory that contains the metric and the $(d-2)$-form $C$ only:[8]

$$\mathbf{E}_a = \iota_a \star (e^c \wedge e^d) \wedge R_{cd} + \frac{1}{2} \left( \iota_a G \wedge \star G + G \wedge \iota_a \star G \right), \tag{9a}$$

$$\mathbf{E} = -d \star G. \tag{9b}$$

We can easily guess the action these equations of motion can be derived from.[9] However, there is a more systematic and direct procedure (often called *Poincaré duality*) that can be used as long as the action depends only on the field strength $d\phi$ and not on the scalar field $\phi$.[10] In these conditions, we can obtain an equivalent action by replacing the scalar field $\phi$ by its 1-form field strength, which we provisionally call $A$, as independent variable as long as we add a Lagrange-multiplier term enforcing the Bianchi identity $dA = 0$. This constraint implies the local existence of $\phi$ and allows us to recover the original scalar equation of motion. Calling $C$ this Lagrange multiplier and defining $G \equiv dC$, the equivalent action takes the form

$$S[e^a, C, A] = \int \left\{ (-1)^{d-1} \star (e^a \wedge e^b) \wedge R_{ab} + \frac{(-1)^d}{2} A \wedge \star A + G \wedge A \right\}. \tag{10}$$

The equation of motion of $A$ is algebraic:

$$A = \star G, \tag{11}$$

and its solution can be used in the above action to get

$$S[e^a, C] = \int \left\{ (-1)^{d-1} \star (e^a \wedge e^b) \wedge R_{ab} + \frac{(-1)^d}{2} G \wedge \star G \right\}, \tag{12}$$

which is the action from which the equations of motion (9a) and (9b) can be derived.

---

[8]We have to take into account that, with our conventions, for a $(k+1)$-form $\omega^{(k+1)}$

$$\star^2 \omega^{(k+1)} = (-1)^{k(d-1)} \omega^{(k+1)}, \tag{7}$$

and also that the canonical normalization of the action of a $k$-form with $(k+1)$-form field strength $\omega^{(k+1)}$ is

$$\frac{(-1)^{dk}}{2} \omega^{(k+1)} \wedge \star \omega^{(k+1)}. \tag{8}$$

[9]As usual in electric-magnetic duality, it is not possible to replace $\phi$ by its dual field $C$ directly in the action since the relation between these variables is non-local, even though the relation between their field strengths is. Substituting $d\phi$ by $\star dC$ directly in the action leads to the wrong sign for the kinetic term of the dual field $C$.

[10]Sometimes it is possible to rewrite an action with explicit dependencies on $\phi$ in such a way that it only depends on $d\phi$. As a general rule, this happens when the action is invariant under constant shifts of $\phi$: $\phi \rightarrow \phi + c$. We will discuss this point in more detail later.

This action is invariant under the gauge transformations Eq. (6). Notice that, by following this procedure, we have obtained the right sign for the kinetic term of $C$.

This dual action is not our real goal, though. We are interested in actions in which the original ("electric") and the dual ("magnetic") variables appear simultaneously. Since this implies a redundancy of degrees of freedom, it is necessary to use the relation between these variables (Eq. (5), in this case) after the equations of motion are derived from the action. Actions which need to be supplemented by constraints in order to derive the equations of motion were called *pseudoactions* in Ref. [34]. Thus, we are interested in pseudoactions which contain both electric and magnetic variables and which give equations of motion equivalent to those of the original theory after the duality relations have been imposed. In the context of $\mathcal{N} = 2, A, B, d = 10$ supergravity (the effective field theories of the type IIA and IIB superstrings), this kind of formulations of the theories were called *democratic* in Ref. [31]. Thus, we are interested in the democratic formulation of the theory given by the original action Eq. (1), which will be described by a pseudoaction.

In this simple case, it is not difficult to see that the pseudoaction we are after, containing $\phi$ and $C$ simultaneously, can be obtained by combining the kinetic terms of $\phi$ and $C$ multiplied by $1/2$,[11]

$$S[e^a, \phi, C] = \int \left\{ (-1)^{d-1} \star (e^a \wedge e^b) \wedge R_{ab} + \tfrac{1}{4} d\phi \wedge \star d\phi + \tfrac{(-1)^d}{4} G \wedge \star G \right\}, \qquad (13)$$

and that it has to be supplemented by the constraint Eq. (5). Indeed, if we use the duality constraint in the equations of motion

$$\mathbf{E}_a = \iota_a \star (e^c \wedge e^d) \wedge R_{cd} + \tfrac{(-1)^d}{4} \left( \iota_a d\phi \wedge \star d\phi + d\phi \wedge \iota_a \star d\phi \right)$$
$$+ \tfrac{1}{4} \left( \iota_a G \wedge \star G + G \wedge \iota_a \star G \right), \qquad (14a)$$

$$\mathbf{E}_\phi = -\tfrac{1}{2} d \star d\phi, \qquad (14b)$$

$$\mathbf{E}_C = -\tfrac{1}{2} d \star G, \qquad (14c)$$

to eliminate $C$, the energy-momentum $(d-1)$-form of $C$ becomes equal to that of $\phi$ and one recovers the Einstein equation (4a) with the right coefficient and the equation of motion of $C$ equation is automatically solved and one is left with the scalar equation of motion with an overall factor of $1/2$. If, instead, one uses the constraint to eliminate $\phi$ one obtains the dual result, *i.e.* the equations of motion (9a) and (9b), the later with an overall factor of $1/2$.

Before ending this section, observe that using the duality relation directly in the democratic pseudoaction does not lead to the original action because the two kinetic terms simply cancel each other.[12] This implies that, if we try to evaluate the action on-shell, since any solution satisfies the duality constraint, the contributions of the kinetic terms will also cancel each other. However, usually, it is the Euclidean action that one is interested in evaluating, not the Lorentzian one. A field and its dual must necessarily have opposite parities and one of them will by multiplied by $i$ when Wick-rotated, its kinetic term acquiring an additional minus sign that will transform the cancellation of the contributions of the dual kinetic terms into its addition.

## 3 Non-linear $\sigma$-models with isometries

The basic dualization procedure used in the previous section will fail when the action cannot be rewritten written in terms of the scalar field strength only. This happens, generically, when the

---

[11]In this case, other, less symmetric combinations of coefficients of the kinetic terms of $\phi$ and $C$ give the same result.

[12]With different coefficients they may not cancel completely, but they will never give the original action back.

scalar field interacts with other fields. We will consider the coupling of scalars to $(p+1)$-form potentials in Section 5 and now we will consider interactions between several scalar fields.

When we have several scalar fields $\phi^x$ in our theory (in absence of scalar potential), the situation becomes more complicated since the scalars can couple non-trivially to the kinetic terms of other scalars. A convenient way to describe all these possibilities in a geometric way is through the non-linear $\sigma$-model formalism in which the scalar fields are interpreted as mappings from spacetime to some "target space" in which they play the role of coordinates. The couplings between scalars and kinetic terms are collected in the $\sigma$-model (or target-space) metric $g_{xy}(\phi)$. The kinetic term (a combination of the kinetic terms of all the scalars and their couplings)

$$\tfrac{1}{2}g_{xy}(\phi)d\phi^x \wedge \star d\phi^y \,, \tag{15}$$

can then be understood as the pull-back of the line element from the target space to spacetime. Scalar field redefinitions can be reinterpreted as general coordinate transformations in the target space.

The action of this system coupled to gravity takes the form

$$S[e^a, \phi^x] = \int \left\{ (-1)^{d-1} \star (e^a \wedge e^b) \wedge R_{ab} + \tfrac{1}{2}g_{xy}(\phi)d\phi^x \wedge \star d\phi^y \right\}, \tag{16}$$

and the equations of motion are

$$\mathbf{E}_a = \iota_a \star (e^c \wedge e^d) \wedge R_{cd} + \tfrac{(-1)^d}{2}g_{xy}\left(\iota_a d\phi^x \wedge \star d\phi^y + d\phi^x \wedge \iota_a \star d\phi^y\right), \tag{17a}$$

$$\mathbf{E}_x = -g_{xy}\left[d \star d\phi^y + \Gamma_{zw}{}^y d\phi^z \wedge \star d\phi^w\right], \tag{17b}$$

where $\Gamma_{zw}{}^y$ are the components of the Christoffel symbols of the target-space metric $g_{xy}$. The scalar equation of motion is, then, the pullback of the geodesic equation in target space.

We would like to dualize the scalars $\phi^x$ into $(d-2)$-form fields. In the single scalar case, we used the fact that the equation of motion $d \star d\phi = 0$ could be understood as a statement on the closedness of certain differential form, $\star d\phi$ that we could locally solve by saying that the differential form is exact $\star d\phi = dC$. In this theory, though, the equations of motion of the scalars only have that form if the Christoffel symbols take a very particular form. Since they are not tensors, this depends very strongly on the coordinates (scalar fields, $\phi^x$) chosen, which complicates the problem of finding out which scalars can be dualized and when.

There is, however, a coordinate-invariant characterization of the scalars that can be dualized based on the following observation: scalar equations of motion which are equivalent to the closedness of a $(d-1)$-form can be interpreted as the conservation law of a $(d-1)$-form current $J$,

$$dJ = 0 \,. \tag{18}$$

If the theory is invariant under global symmetries acting on the scalar fields, Noether's theorem ensures that there will be as many conserved currents as symmetries. The on-shell conservation laws of these currents will be combinations of some of the equations of motion of the scalar fields that can be used to dualize them.

In order to characterize the scalar symmetries of the theory we denote by

$$\delta_A \phi^x = k_A{}^x(\phi), \tag{19}$$

their infinitesimal generators. The indices $A$ label the independent symmetries,[13] and $k_A{}^x(\phi)$ are some given (not arbitrary) functions of the scalar fields, as the symmetries we are considering are global (not local) and take the form

$$\delta\phi^x = \alpha^A k_A{}^x(\phi), \tag{20}$$

for constant, infinitesimal parameters $\alpha^A$.

---

[13]That is: they take values in the adjoint representation of the Lie algebra of the symmetry group $G$.

It is not hard to see that the action Eq. (16) is invariant under the transformations Eq. (19) if and only if the $k_A{}^x(\phi)$ are Killing vectors of the target-space metric, *i.e.* if

$$\nabla_{(x|}k_{A|y)} = 0\,, \tag{21}$$

where $\nabla_x$ is the target-space covariant derivative with the connection $\Gamma_{xy}{}^z$ and $k_{Ax} = k_A{}^y g_{yx}$. The associated Noether current $(d-1)$-forms are given by

$$J_A = \star\hat{k}_A\,, \tag{22}$$

where $\hat{k}_A$ is the pullback of the 1-forms dual to the Killing vectors

$$\hat{k}_A \equiv k_A{}^x g_{xy} d\phi^y\,. \tag{23}$$

Furthermore, it is not difficult to see, using the Killing equation, that

$$k_A{}^x \mathbf{E}_x = -dJ_A\,, \tag{24}$$

which establishes the relation between the scalar equations of motion and the on-shell conservation of the Noether currents we were looking for.

The above conservation laws suggest that we may try to define dual $(d-2)$-forms $C_A$ with field strengths $G_A$ associated the conserved currents via

$$G_A \equiv dC_A = J_A\,. \tag{25}$$

Since the currents $J_A$ only transform under global $G$ transformations, $G_A$ must be gauge invariant and the $(d-2)$-forms only transform under gauge transformations

$$\delta_\Sigma C_A = d\Sigma_A\,. \tag{26}$$

The currents $J_A$ do not occur explicitly in the action and, therefore, it is not clear how one can use the Poincaré duality procedure. When the Killing vectors $k_A{}^x$ commute, though, it is possible to use coordinates adapted to all the isometries.[14] In this adapted coordinate system the target-space metric $g_{xy}$ is independent of the scalars associated to the Killing vectors and they can be Poincaré-dualized in the standard fashion because those scalars only occur through their field strengths. We are going to consider this particular case first.

## 3.1 Non-linear $\sigma$-models with commuting isometries

In this case we can use the machinery and notation (hats for original fields) of Kaluza-Klein dimensional reductions for the target space metric. We choose coordinates adapted to all the commuting isometries (all the Killing vectors $k_A{}^x$) to be considered splitting the coordinate indices into those related to the isometries, $A$, and the rest, $m$, $\{x\} = \{A, m\}$ and using the notation $\{\hat{\phi}^x\} = \{\varphi^A, \phi^m\}$. In these coordinates, the components of the Killing vectors are $k_A{}^x = \delta_A{}^x$ (i.e. $k_A{}^B = \delta_A{}^B$, $k_A{}^m = 0$).

By definition, the target-space metric only depends on the scalar fields $\phi^m$ and its components and those of its inverse can be written in the form

$$
\begin{aligned}
(\hat{g}_{xy}) &= \begin{pmatrix} g_{AB} & g_{AC}A^C{}_n \\[2mm] g_{BC}A^C{}_m & g_{mn} + A^A{}_m A^B{}_n g_{AB} \end{pmatrix}\,, \\[3mm]
(\hat{g}^{xy}) &= \begin{pmatrix} g^{AB} + A^A{}_m A^B{}_n g^{mn} & -A^A{}_p g^{pn} \\[2mm] -A^B{}_p g^{pm} & g^{mn} \end{pmatrix}\,,
\end{aligned}
\tag{27}
$$

---

[14]We can also restrict ourselves to an Abelian subgroup of the isometry group of the target space metric.

where

$$g^{AB}g_{BC} = \delta^A{}_C\,, \qquad g^{mn}g_{np} = \delta^m{}_p\,. \tag{28}$$

We stress that all the target-space fields $g_{AB}, g_{mn}$ and $A^A{}_m$ are independent of the scalars $\varphi^A$.

In terms of these new variables (actually, combinations of scalar fields), the action Eq. (16) takes the form

$$S[e^a, \varphi^A, \phi^m] = \int \left\{ (-1)^{d-1} \star (e^a \wedge e^b) \wedge R_{ab} + \tfrac{1}{2} g_{AB} \mathcal{D}\varphi^A \wedge \star \mathcal{D}\varphi^B + \tfrac{1}{2} g_{mn} d\phi^m \wedge \star d\phi^n \right\}, \tag{29}$$

where we have defined

$$\mathcal{D}\varphi^A \equiv d\varphi^A + A^A{}_m d\phi^m\,. \tag{30}$$

The equations of motion take now the form

$$\mathbf{E}_a = \iota_a \star (e^c \wedge e^d) \wedge R_{cd} + \tfrac{(-1)^d}{2} g_{AB} \left( \iota_a \mathcal{D}\varphi^A \wedge \star \mathcal{D}\varphi^B + \mathcal{D}\varphi^A \wedge \iota_a \star \mathcal{D}\varphi^B \right)$$
$$+ \tfrac{(-1)^d}{2} g_{mn} \left( \iota_a d\phi^m \wedge \star d\phi^n + d\phi^m \wedge \iota_a \star d\phi^n \right), \tag{31a}$$

$$\mathbf{E}_A = -d\left[ g_{AB} \star \mathcal{D}\varphi^B \right], \tag{31b}$$

$$\mathbf{E}_m = -g_{mn}\left[ d \star d\phi^n + \Gamma_{pq}{}^n d\phi^p \wedge \star d\phi^q \right] + \tfrac{1}{2} \frac{\partial g_{AB}}{\partial \phi^m} \mathcal{D}\varphi^A \wedge \star \mathcal{D}\varphi^B$$
$$+ g_{AB} F^A{}_{mn} d\phi^n \wedge \star \mathcal{D}\varphi^B + A^A{}_m \mathbf{E}_A\,. \tag{31c}$$

The equations of motion of the scalars $\varphi^A$ can be understood as the expression of the conservation of the Noether currents associated to the invariance of the action under the constant shifts generated by the Killing vectors because the currents $J_A$ are given by

$$J_A \equiv g_{AB} \star \mathcal{D}\varphi^B \quad \Rightarrow \quad \mathbf{E}_A = -dJ_A\,. \tag{32}$$

Then, we can solve locally the equations of motion of those scalars by introducing the dual $(d-2)$-forms $C_A$:

$$G_A \equiv dC_A = J_A\,. \tag{33}$$

The field strengths $G_A$ are invariant under gauge transformations of the form

$$\delta_\Sigma C_A = d\Sigma_A\,, \tag{34}$$

where the $\Sigma_A$ are $(d-3)$-forms.

The duality relation Eq. (33) together with the definition of the currents $J_A$ can be used to express the field strengths of the scalars $\varphi^A$ in terms of the dual $(d-2)$-forms $C_A$ (and the rest of the scalars)

$$d\varphi^A = g^{AB} \star G_B - A^A{}_m d\phi^m\,. \tag{35}$$

Then, the Bianchi identity of these field strengths, $d^2\varphi^A = 0$) gives the equations of motion of $(d-2)$-forms $C_A$:

$$d\left( g^{AB} \star G_B \right) - F^A = 0\,, \tag{36}$$

where

$$F^A \equiv \tfrac{1}{2} F^A{}_{mn} d\phi^m \wedge d\phi^n\,, \qquad \text{with} \qquad F^A{}_{mn} \equiv 2\partial_{[m}A^A{}_{n]}\,. \tag{37}$$

The scalar fields $\varphi^A$ can be completely eliminated from the action by standard Poincaré dualization and the result is an action that contains the field variables $C_A$ and $\phi^m$ (which we do not know how to dualize) and which is invariant up to a total derivative under the $\delta_\Sigma$ gauge transformations defined in Eq. (34)

$$S[e^a, C_A, \phi^m] = \int \left\{ (-1)^{d-1} \star (e^a \wedge e^b) \wedge R_{ab} + \tfrac{(-1)^d}{2} g^{AB} G_A \wedge \star G_B + C_A \wedge F^A \right.$$
$$\left. + \tfrac{1}{2} g_{mn} d\phi^m \wedge \star d\phi^n \right\}. \tag{38}$$

The equations of motion that follow from this action are

$$\mathbf{E}_a = \iota_a \star (e^c \wedge e^d) \wedge R_{cd} + \tfrac{1}{2} g^{AB} \left( \iota_a G_A \wedge \star G_B + (-1)^d G_A \wedge \iota_a \star G_B \right)$$
$$+ \tfrac{(-1)^d}{2} g_{mn} (\iota_a d\phi^m \wedge \star d\phi^n + d\phi^m \wedge \iota_a \star d\phi^n), \tag{39a}$$

$$\mathbf{E}^A = -d(g^{AB} \star G_B) + F^A, \tag{39b}$$

$$\mathbf{E}_m = -g_{mn}[d \star d\phi^n + \Gamma_{pq}{}^n d\phi^p \wedge \star d\phi^q] + \tfrac{(-1)^d}{2} \frac{\partial g^{AB}}{\partial \phi^m} G_A \wedge \star G_B$$
$$- 2(-1)^{d-1} G_A \wedge F^A{}_{nm} d\phi^n, \tag{39c}$$

are completely equivalent to those of the original fields upon use of the duality relation Eq. (33).[15] Furthermore, we can construct a democratic action by simply adding the original kinetic term of the $\varphi^A$s to this action, changing the normalization of the kinetic terms to get the right normalization of the energy-momentum tensor in the Einstein equation (the topological term $C_A \wedge F^A$ does not contribute to it):

$$S[e^a, C_A, \varphi^S, \phi^m] = \int \left\{ (-1)^{d-1} \star (e^a \wedge e^b) \wedge R_{ab} + \tfrac{1}{4} g_{AB} \mathcal{D}\varphi^A \wedge \star \mathcal{D}\varphi^B \right.$$
$$\left. + \tfrac{(-1)^d}{4} g^{AB} G_A \wedge \star G_B + \tfrac{1}{2} C_A \wedge F^A + \tfrac{1}{2} g_{mn} d\phi^m \wedge \star d\phi^n \right\}. \tag{40}$$

The equations of motion that follow from this action are

$$\mathbf{E}_a = \iota_a \star (e^c \wedge e^d) \wedge R_{cd} + \tfrac{1}{4} g^{AB} \left( \iota_a G_A \wedge \star G_B + (-1)^d G_A \wedge \iota_a \star G_B \right)$$
$$+ \tfrac{(-1)^d}{2} g_{mn} (\iota_a d\phi^m \wedge \star d\phi^n + d\phi^m \wedge \iota_a \star d\phi^n)$$
$$+ \tfrac{(-1)^d}{4} g_{AB} (\iota_a \mathcal{D}\varphi^A \wedge \star \mathcal{D}\varphi^B + \mathcal{D}\varphi^A \wedge \iota_a \star \mathcal{D}\varphi^B), \tag{41a}$$

$$\mathbf{E}_A = -\tfrac{1}{2} d(g_{AB} \star \mathcal{D}\varphi^B), \tag{41b}$$

$$\mathbf{E}^A = -\tfrac{1}{2} d(g^{AB} \star G_B) + \tfrac{1}{2} F^A, \tag{41c}$$

$$\mathbf{E}_m = -g_{mn}[d \star d\phi^n + \Gamma_{pq}{}^n d\phi^p \wedge \star d\phi^q] + \frac{1}{4} \frac{\partial g_{AB}}{\partial \phi^m} \mathcal{D}\varphi^A \wedge \star \mathcal{D}\varphi^B$$

$$+ \tfrac{(-1)^d}{4} \frac{\partial g^{AB}}{\partial \phi^m} G_A \wedge \star G_B + g_{AB} F^A{}_{mn} d\phi^n \wedge \star \mathcal{D}\varphi^B$$

$$- (-1)^{d-1} G_A \wedge F^A{}_{nm} d\phi^n. \tag{41d}$$

This is a simple case in which the scalars $\varphi^A$ can be completely replaced by its dual $(d-2)$-forms $C_A$. Still, we have learned that it is necessary to include the topological term $C_A \wedge F^A$ in the dual action.

It is not clear at all how to dualize the rest of the scalars, if this is possible at all. On general grounds we expect the scalars related to symmetries to be "dualizable" because their equations of motion are related to the conservation of certain Noether currents and then we can dualize on shell those equations. If there are enough symmetries, we may be able to dualize all the scalars, at least in the sense of being able to define the $(d-2)$-form potentials dual to them. However, when the isometries do not commute, we cannot use coordinates adapted to all the isometries and we cannot use the Poincaré dualization procedure and, any putative action containing the dual fields should also include the original scalar fields. On the other hand, if we do use all the currents (all the isometries) the dual $(d-2)$-form potentials will not fill a linear representation of the symmetry group and the invariance of the theory containing the dual fields under this group will in general be broken.

---

[15]The Bianchi identity of the target-space 2-form field strengths $F^A$ occur in the equation of motion of the $\phi^m$ and explains why the term $C_A \wedge F^A$ does not contribute to them.

We do not know how to solve this problem in general. However, in the case in which the target space is a Riemannian symmetric manifold, inspired by the form of the action that we have just constructed (the necessity of the topological term $C_A \wedge F^A$), we have found a way to construct a democratic action which manifestly preserves all the symmetries of the original one. We describe this construction in the next section.

## 4  Dualization of Riemannian symmetric $\sigma$-models

In this section we are interested in the case in which the target-space metric $g_{xy}(\phi)$ of the non-linear $\sigma$-model action Eq. (16) is that of a $G/H$ coset space which is a Riemannian symmetric space.[16] In particular, we are going to assume that $g_{xy}(\phi)$ has been constructed using the restriction of the Killing metric of $G$, $g_{AB}$, to the horizontal space,[17] $g_{mn}$ and the horizontal components of the Maurer-Cartan 1-form $v^m = v^m{}_x d\phi^x$ as Vielbeins:

$$g_{xy} = g_{mn} v^m{}_x v^n{}_y \,. \tag{42}$$

Thus, $g_{xy}$ admits, at least,[18] $\dim G$ Killing vectors, $k_A{}^x$, which generate as many global symmetries of the action Eq. (19). Associated to them, there are $\dim G$ closed $(d-1)$-form currents, $J_A$, of the form Eq. (22) and, by construction, there are more conserved currents ($\dim G$) than physical scalars ($\dim G$-$\dim H$). However, as noticed in Ref. [39], only if we use all of them will the whole global symmetry group, $G$, be preserved. Therefore, we must define $\dim G$ $(d-2)$-forms $C_A$ and their respective field strengths $G_A$, through Eq. (25) and we must use all of them in the action, but we must find a way to make $\dim H$ of the $(d-2)$-forms $C_A$ non-dynamical.

On the other hand, as we have discussed, it is clear that it is impossible to construct an equivalent action in which only the dual $(d-2)$-forms $C_A$, and not the scalar fields $\phi^x$ occur. The best we can hope for is a democratic pseudoaction.

Taking into account all this and the discussions in Ref. [39], we propose the following democratic action for all these fields:

$$S_{\text{Dem}}[e^a, \phi^x, C_A] = \int \Big\{ (-1)^{d-1} \star (e^a \wedge e^b) \wedge R_{ab} + \tfrac{1}{4} g_{xy} d\phi^x \wedge d\phi^y$$
$$+ \tfrac{(-1)^d}{4} \mathfrak{M}^{AB} G_A \wedge \star G_B - \tfrac{(-1)^d}{2} g^{AB} G_A \wedge \hat{k}_B \Big\} \,, \tag{43}$$

where the $\dim G \times \dim G$ matrix $\mathfrak{M}^{AB}$ is defined as

$$\mathfrak{M}^{AB} = g^{AC} g^{BD} k_C{}^x k_D{}^y g_{xy} \,, \tag{44}$$

and where the equations of motion are meant to be supplemented by the duality relations Eqs. (25).

It is not difficult to prove that the matrix $\mathfrak{M}^{AB}$ has $\dim G$-$\dim H$ eigenvectors so that $\text{rank}\,\mathfrak{M} = \dim G$-$\dim H$.[19] This means that in the above action there are $\dim H$ combinations of

---

[16]In this section we are going to use, with minimal changes, the notation and conventions of Refs. [14, 39] to which we refer for further references and details on symmetric $\sigma$-models.

[17]This vector space, $\mathfrak{t}$ is the complement of the Lie subalgebra $\mathfrak{h}$ of H in $\mathfrak{g}$ (the Lie algebra of $G$), that is, $\mathfrak{g} = \mathfrak{h} \oplus \mathfrak{t}$. We use indices $A, B, \cdots = 1, \ldots, \dim G$ to label the adjoint representation of $G$, indices $i, j, \ldots = 1, \ldots, \dim H$ to label that of $H$ and $m, n, \ldots = 1, \ldots, \dim G$-$\dim H$ to label a basis of $\mathfrak{t}$. The scalars are labeled by $x, y, \ldots = 1, \ldots, \dim G - \dim H$.

[18]We will ignore, for the sake of simplicity, any other Killing vectors of $g_{xy}$.

[19]The eigenvectors are, precisely, the $\dim H$ *momentum maps* $P_A{}^i = \Gamma_{\text{Adj}}(u^{-1})^i{}_A$ [39]: taking into account that the Killing vectors are given by $k_A{}^m = -\Gamma_{\text{Adj}}(u^{-1})^m{}_A$ and that the Killing metric is $g$-invariant and block-diagonal,

$$\mathfrak{M}^{AB} P_B{}^i = -g^{AC} k_C{}^m g_{mn} \Gamma_{\text{Adj}}(u^{-1})^n{}_A g^{BD} \Gamma_{\text{Adj}}(u^{-1})^i{}_A = -g^{AC} k_C{}^m g_{mn} g^{ni} = 0 \,. \tag{45}$$

the magnetic field strengths $G_A$ which do not occur in the kinetic term, which is what we need in order not to have too many dynamical fields.

We are going to show that the equations of motion that follow from the above democratic action are those of the original $\sigma$-model upon use of the duality relations Eqs. (25).

Observe that the original kinetic term for the scalar fields can be rewritten in terms of the Noether 1-form currents using the matrix $\mathfrak{M}$ as

$$\mathfrak{M}^{AB} \star J_A \wedge J_B = g^{AC} g^{BD} k_C{}^x k_D{}^y g_{xy} k_{Az} k_{Bw} d\phi^z \wedge \star d\phi^w = g_{xy} d\phi^x \wedge \star d\phi^y \,, \qquad (47)$$

by virtue of the property

$$g^{AB} k_A{}^m k_B{}^n = g^{mn} \quad \Rightarrow \quad g^{AB} k_A{}^x k_B{}^y = g^{xy} \,. \qquad (48)$$

Observe that, then,

$$\mathfrak{M}^{AB} k_{Ax} k_{By} = g^{AB} k_{Ax} k_{By} = g_{xy} \,. \qquad (49)$$

The Einstein equations that follow from the democratic action are

$$\mathbf{E}_a = \iota_a \star (e^c \wedge e^d) \wedge R_{cd} + \frac{(-1)^d}{4} g_{xy} \left( \iota_a d\phi^x \star d\phi^y + d\phi^x \wedge \iota_a \star d\phi^y \right)$$
$$+ \tfrac{1}{4} \mathfrak{M}^{AB} \left( \iota_a G_A \wedge \star G_B + (-1)^d G_A \iota_a \star G_B \right) \,. \qquad (50)$$

It is enough to consider the last term (the energy-momentum $(d-1)$-form of the dual $(d-2)$-forms). Using the duality relations Eqs. (25) and the property Eq. (46), that term takes the form

$$\tfrac{1}{4} \mathfrak{M}^{AB} \left( \iota_a \star \hat{k}_A \wedge \hat{k}_B + (-1)^d \star \hat{k}_A \iota_a \hat{k}_B \right) = \frac{(-1)^d}{4} \mathfrak{M}^{AB} \left( \hat{k}_A \wedge \iota_a \star \hat{k}_B + \iota_a \hat{k}_A \hat{k}_B \right)$$
$$= \frac{(-1)^d}{4} \mathfrak{M}^{AB} k_{Ax} k_{By} \left( d\phi^x \wedge \iota_a \star d\phi^y + \iota_a d\phi^x \star d\phi^y \right)$$
$$= \frac{(-1)^d}{4} g_{xy} \left( d\phi^x \wedge \iota_a \star d\phi^y + \iota_a d\phi^x \star d\phi^y \right) \,, \qquad (51)$$

by virtue of Eq. (49) and it can be added to the energy-momentum $(d-1)$-form of the scalars to recover the energy-momentum tensor of the scalars in the original theory.

The equations of motion of the scalars are, after use of the Killing equation

$$\mathbf{E}_x = -\tfrac{1}{2} g_{xy} \left\{ d \star d\phi^y + \Gamma_{zw}{}^y d\phi^z \wedge \star d\phi^w \right\} + \frac{(-1)^d}{4} \partial_x \mathfrak{M}^{AB} G_A \wedge \star G_B$$
$$- \tfrac{1}{2} g^{AB} k_{Ax} dG_B + (-1)^{d+1} g^{AB} \nabla_x k_{Ay} G_B \wedge d\phi^y \,. \qquad (52)$$

Let us consider the last term first. Using the duality relations Eqs. (25)

$$(-1)^{d+1} g^{AB} \nabla_x k_{Ay} k_{Bz} \star d\phi^z \wedge d\phi^y = \frac{(-1)^{d+1}}{2} \nabla_x g_{yz} \star d\phi^z \wedge d\phi^y = 0 \,, \qquad (53)$$

since we are using the target-space Levi-Civita connection. The second term can be put in the form

$$\frac{(-1)^d}{2} g^{AC} g^{BD} \nabla_x k_C{}^y k_D{}^y k_{Az} k_{Bw} \star d\phi^z \wedge d\phi^w = \frac{(-1)^d}{2} g^{AC} \nabla_x k_{Cw} k_{Az} \star d\phi^z \wedge d\phi^w = 0 \,, \qquad (54)$$

for the same reason.

---

On the other hand, using the same properties we can show that

$$\mathfrak{M}^{AB} k_B{}^m = g^{AB} k_B{}^m \,. \qquad (46)$$

These results are compatible because in $G/H$ only $\dim G - \dim H$ vectors are linearly independent at any given point.

The third (next to last) term in Eq. (52) vanishes on account of the Bianchi identity of $G_A$, which is related to the equation of motion of the scalars. Thus, instead of throwing it away, we are going to use the duality relation and Eq. (24) in it

$$
\begin{aligned}
-\tfrac{1}{2} g^{AB} k_{Ax} d G_B &= -\tfrac{1}{2} g^{AB} k_{Ax} d J_B \\
&= -\tfrac{1}{2} g^{AB} k_{Ax} k_{B\,y} \{ d \star d\phi^y + \Gamma_{zw}{}^y d\phi^z \wedge \star d\phi^w \} \\
&= -\tfrac{1}{2} g_{xy} \{ d \star d\phi^y + \Gamma_{zw}{}^y d\phi^z \wedge \star d\phi^w \} \,,
\end{aligned}
\tag{55}
$$

and we recover the original equation of motion of the scalars with identical normalization.

Finally, the equations of motion of the $(d-2)$-form potentials $C_A$

$$
\mathbf{E}^A = \tfrac{1}{2} d \left[ \mathfrak{M}^{AB} \star G_B - g^{AB} \hat{k}_B \right] ,
\tag{56}
$$

are solved automatically by the duality relation upon use of the properties of the matrix $\mathfrak{M}^{AB}$.

The scalars of all the maximal and half-maximal supergravities parametrize a Riemannian symmetric $\sigma$-model. However, in all those theories they are also coupled to other fields. In the next section we consider the coupling to $(p+1)$-form potentials as a toy model since "real" supergravities usually have several of these with different ranks and with Chern-Simons terms in the action and field strengths.

# 5 Dualization of Riemannian symmetric $\sigma$-models coupled to $(p+1)$-forms

The next step consists in the coupling of a Riemannian symmetric $\sigma$-model to a set of $(p+1)$-form potentials (the fields $p$-branes naturally couple to)

$$
A^\Lambda = \frac{1}{(p+1)!} A^\Lambda{}_{\mu_1 \cdots \mu_{p+1}} dx^{\mu_1} \wedge \cdots \wedge dx^{\mu_{p+1}} \,,
\tag{57}
$$

with $(p+2)$-form field strengths

$$
F^\Lambda = dA^\Lambda \,,
\tag{58}
$$

invariant under the gauge transformations

$$
\delta_\chi A^\Lambda = d\chi^\Lambda \,,
\tag{59}
$$

where each $\chi^\Lambda$ is an arbitrary $p$-form.

In arbitrary dimension $d$, the action that describes this coupling takes the generic form

$$
S[e^a, A^\Lambda, \phi^x] = \int \left\{ (-1)^{d-1} \star (e^a \wedge e^b) \wedge R_{ab} + \tfrac{1}{2} g_{xy} d\phi^x \wedge \star d\phi^y - \frac{(-1)^{(p+1)d}}{2} I_{\Lambda\Sigma} F^\Lambda \wedge \star F^\Sigma \right\} , \tag{60}
$$

$I_{\Lambda\Sigma}$ being a symmetric and negative-definite scalar-dependent matrix.

The equations of motion that follow from this action are

$$
\begin{aligned}
\mathbf{E}_a = {}& \iota_a \star (e^c \wedge e^d) \wedge R_{cd} + \frac{(-1)^d}{2} g_{xy} \left( \iota_a d\phi^x \wedge \star d\phi^y + d\phi^x \wedge \iota_a \star d\phi^y \right) \\
& + \frac{(-1)^{(p+1)d}}{2} I_{\Lambda\Sigma} \left( \iota_a F^\Lambda \wedge \star F^\Sigma + (-1)^{p+1} F^\Lambda \wedge \iota_a \star F^\Sigma \right) ,
\end{aligned}
\tag{61a}
$$

$$
\mathbf{E}_x = -g_{xy} \left[ d \star d\phi^y + \Gamma_{zw}{}^y d\phi^z \wedge \star d\phi^w \right] - \frac{(-1)^{(p+1)d}}{2} \partial_x I_{\Lambda\Sigma} F^\Lambda \wedge \star F^\Sigma \,,
\tag{61b}
$$

$$
\mathbf{E}_\Lambda = d \left( I_{\Lambda\Omega} \star F^\Omega \right) .
\tag{61c}
$$

For a non-constant kinetic matrix $I_{\Lambda\Sigma}$ the action is invariant under all the transformations of the scalars Eq. (19) associated to the symmetries of the $\sigma$-model, if the $(p+1)$-forms also transform according to

$$\delta_A A^\Lambda = T_A{}^\Lambda{}_\Sigma A^\Sigma \,, \tag{62}$$

for some matrices $T_A$, and the kinetic matrix $I_{\Lambda\Sigma}$ satisfies the property

$$\delta_A I_{\Lambda\Sigma} = -\pounds_{k_A} I_{\Lambda\Sigma} = -2T_A{}^\Omega{}_{(\Lambda} I_{\Sigma)\Omega} \,. \tag{63}$$

The Noether current $(d-1)$-forms associated to these symmetries are

$$J_A = \star \hat{k}_A + (-1)^{(p+1)d} T_A{}^\Lambda{}_\Sigma A^\Sigma \wedge \left(I_{\Lambda\Omega} \star F^\Omega\right) \,. \tag{64}$$

The first term of this current is invariant under the gauge transformations Eq. (59) but the second is not: it transforms into a total derivative on-shell. Thus, if we try to dualize $J_A$ using its conservation law $dJ_A = 0$

$$\star \hat{k}_A + (-1)^{(p+1)d} T_A{}^\Lambda{}_\Sigma A^\Sigma \wedge \left(I_{\Lambda\Omega} \star F^\Omega\right) \equiv d\tilde{C}_A \,, \tag{65}$$

the right definition for a gauge-invariant field strength is

$$\star \hat{k}_A = d\tilde{C}_A - (-1)^{(p+1)d} T_A{}^\Lambda{}_\Sigma A^\Sigma \wedge \left(I_{\Lambda\Omega} \star F^\Omega\right) \equiv G_A \,, \tag{66}$$

and the total derivative generated by the gauge transformations of the $(p+1)$-form potentials $A^\Lambda$ must be absorbed by a gauge transformation of the $(d-2)$-form potentials that we will described shortly.

The Chern-Simons term in the field strength is unusual but can be transformed using the dual of the $(p+1)$-form potentials: their equations of motion $\mathbf{E}_\Lambda = 0$ can be locally solved with the introduction of the dual $(\tilde{p}+1)$-forms $\tilde{A}_\Lambda$, with $\tilde{p} \equiv d-p-4$:

$$I_{\Lambda\Omega} \star F^\Omega \equiv d\tilde{A}_\Lambda \equiv \tilde{F}_\Lambda \,. \tag{67}$$

The field strengths $\tilde{F}_\Lambda$ are invariant under the dual gauge transformations

$$\delta_{\tilde{\chi}} \tilde{A}_\Lambda = d\tilde{\chi}_\Lambda \,. \tag{68}$$

Using this definition, we can write

$$G_A = d\tilde{C}_A - (-1)^{(p+1)d} T_A{}^\Lambda{}_\Sigma A^\Sigma \wedge \tilde{F}_\Lambda \,, \tag{69}$$

and, integrating by parts in order to get a more symmetric expression, we arrive at the final definition of the $(d-2)$-form potentials and their field strengths dual to the scalars:

$$G_A \equiv dC_A - \frac{(-1)^{(p+1)d}}{2} T_A{}^\Lambda{}_\Sigma \left(A^\Sigma \wedge \tilde{F}_\Lambda + (-1)^{p(d+1)} \tilde{A}_\Lambda \wedge F^\Sigma\right) = \star \hat{k}_A \,. \tag{70}$$

The gauge invariance of the field strengths $G_A$ implies the following gauge transformations of the $(d-2)$-form potentials

$$\delta C_A = d\Sigma_A + \frac{(-1)^{(p+1)d}}{2} T_A{}^\Lambda{}_\Sigma \left(\chi^\Sigma \wedge \tilde{F}_\Lambda + (-1)^{p(d+1)} \tilde{\chi}_\Lambda \wedge F^\Sigma\right) \,. \tag{71}$$

Observe that in this theory

$$k_A{}^x \mathbf{E}_x = -dJ_A + (-1)^{(p+1)(d+1)} T_A{}^\Lambda{}_\Sigma A^\Sigma \wedge \mathbf{E}_\Lambda \,. \tag{72}$$

Based on our previous experience, we propose the following gauge- and $G$-invariant democratic action for this theory:

$$S_{\text{Dem}}[e^a, A^\Lambda, \tilde{A}_\Lambda, \phi^x, C_A] = \int \left\{ (-1)^{d-1} \star (e^a \wedge e^b) \wedge R_{ab} + \tfrac{1}{4} g_{xy} d\phi^x \wedge d\phi^y \right.$$
$$- \frac{(-1)^{(p+1)d}}{4} I_{\Lambda\Sigma} F^\Lambda \wedge \star F^\Sigma - \frac{(-1)^{(\tilde{p}+1)d}}{4} I^{\Lambda\Sigma} \tilde{F}_\Lambda \wedge \star \tilde{F}_\Sigma$$
$$\left. + \frac{(-1)^d}{4} \mathfrak{M}^{AB} G_A \wedge \star G_B - \frac{(-1)^d}{2} g^{AB} G_A \wedge \hat{k}_B \right\}, \quad (73)$$

where

$$I^{\Lambda\Omega} I_{\Omega\Sigma} = \delta^\Lambda{}_\Sigma, \quad (74)$$

where we must use the duality relations

$$I_{\Lambda\Omega} \star F^\Omega = \tilde{F}_\Lambda, \quad (75a)$$

$$\star \hat{k}_A = G_A, \quad (75b)$$

in the equations of motion in order to recover those of the original theory,[20] as it can easily be checked.

Coupling the scalars to more potentials of different ranks should only involve more Chern-Simons terms in the definition of the dual field strengths $G_A$. However, there can be additional complications, as we will see in the case of the $\mathcal{N} = 2B, d = 10$ theory.

On the other hand, when $d = 2(p + 2)$ new couplings between the scalars and $(p + 1)$-form potentials are possible. This is a specially interesting case because it includes all the $\mathcal{N} \geq 3, d = 4$ ungauged supergravities and because some of the symmetries of the theory (electric-magnetic dualities) are realized as symmetries of the equations of motion only. We consider it next.

## 5.1 The $d = 2(p + 2)$ case and electric-magnetic dualities

### 5.1.1 The theory and its dualities

When $d = 2(p + 2)$ it is possible to add a gauge-invariant topological (metric-independent) term to the action Eq. (60), which takes the generic form

$$S[e^a, A^\Lambda, \phi^x] = \int \left\{ -\star (e^a \wedge e^b) \wedge R_{ab} + \tfrac{1}{2} g_{xy} d\phi^x \wedge \star d\phi^y - \tfrac{1}{2} I_{\Sigma\Lambda} F^\Sigma \wedge \star F^\Lambda - \tfrac{1}{2} R_{\Sigma\Lambda} F^\Sigma \wedge F^\Lambda \right\}, \quad (76)$$

where the new matrix $R_{\Sigma\Lambda}$ also depends on the scalar fields.

While $I_{\Lambda\Sigma}$ is always symmetric (and, conventionally, negative-definite) the symmetry properties of $R_{\Sigma\Lambda}$ depend on the dimension $d$:[21]

$$R^T = -\xi^2 R, \qquad \xi^2 = (-1)^{p+1}. \quad (78)$$

---

[20]Those of the $(p + 1)$-form potentials appear with a factor of $1/2$.

[21]One should also take into account that, for $(p + 2)$-forms in $d = 2(p + 2)$ dimensions

$$\star^2 F^\Lambda = \xi^2 F^\Lambda, \quad (77a)$$
$$F^\Lambda \wedge F^\Sigma = -\xi^2 F^\Sigma \wedge F^\Lambda, \quad (77b)$$
$$\star F^\Lambda \wedge \star F^\Sigma = -F^\Lambda \wedge F^\Sigma. \quad (77c)$$

The equations of motion that follow from this action are

$$\mathbf{E}_c = \iota_c \star \left(e^a \wedge e^b\right) \wedge R_{ab} + \tfrac{1}{2} g_{xy} \left(\iota_c d\phi^x \wedge \star d\phi^y + d\phi^x \wedge \iota_c \star d\phi^y\right)$$
$$- \tfrac{1}{2} I_{\Sigma\Lambda} \left(\iota_c F^\Sigma \wedge \star F^\Lambda + \xi^2 F^\Sigma \wedge \iota_c \star F^\Lambda\right), \tag{79a}$$

$$\mathbf{E}_z = -g_{zw}\left[d \star d\phi^w + \Gamma_{xy}{}^w d\phi^x \wedge \star d\phi^y\right] - \tfrac{1}{2}\partial_z I_{\Sigma\Lambda} F^\Sigma \wedge \star F^\Lambda - \tfrac{1}{2}\partial_z R_{\Sigma\Lambda} F^\Sigma \wedge F^\Lambda, \tag{79b}$$

$$\mathbf{E}_\Sigma = d\left(I_{\Sigma\Lambda} \star F^\Lambda + R_{\Sigma\Lambda} F^\Lambda\right), \tag{79c}$$

The equations of motion of the $(p+1)$-forms $A^\Lambda$ can be locally solved by introducing dual $(p+1)$-forms $A_\Lambda$ such that

$$I_{\Sigma\Lambda} \star F^\Lambda + R_{\Sigma\Lambda} F^\Lambda = dA_\Lambda \equiv F_\Lambda, \tag{80}$$

where $F_\Lambda$ are the associated $(p+2)$-form field strengths. it is, then, natural, to introduce

$$\left(A^M\right) \equiv \begin{pmatrix} A^\Lambda \\ A_\Lambda \end{pmatrix}, \qquad \left(F^M\right) \equiv d\left(A^M\right) \equiv \begin{pmatrix} F^\Lambda \\ F_\Lambda \end{pmatrix}. \tag{81}$$

The dual field strengths $F_\Lambda$ have been defined in terms of the original ones $F^\Lambda$ and the scalars. Therefore, it is not surprising that $F^M$ satisfies the so-called *twisted self-duality constraint*

$$\star F^M = \xi^2 \Omega^{MN} \mathcal{M}_{NP} F^P, \tag{82}$$

where we have defined

$$\Omega \equiv (\Omega_{MN}) \equiv \begin{pmatrix} 0 & \delta_\Lambda{}^\Sigma \\ \xi^2 \delta^\Lambda{}_\Sigma & 0 \end{pmatrix}, \quad \Omega^{-1} \equiv \left(\Omega^{MN}\right) \equiv \begin{pmatrix} 0 & \xi^2 \delta^\Lambda{}_\Sigma \\ \delta_\Lambda{}^\Sigma & 0 \end{pmatrix} = \xi^2 \Omega, \tag{83}$$

which, for $\xi^2 = -1$ is the $\mathrm{Sp}(2n, \mathbb{R})$ metric and for $\xi^2 = +1$ is the off-diagonal $\mathrm{O}(n,n)$ metric, and the symmetric scalar matrix

$$\mathcal{M} = (\mathcal{M}_{MN}) = \begin{pmatrix} I_{\Lambda\Sigma} - \xi^2 R_{\Lambda\Gamma} I^{\Gamma\Omega} R_{\Omega\Sigma} & \xi^2 R_{\Lambda\Gamma} I^{\Gamma\Sigma} \\ -I^{\Lambda\Gamma} R_{\Gamma\Sigma} & I^{\Lambda\Sigma} \end{pmatrix} = \begin{pmatrix} I - \xi^2 R I^{-1} R & \xi^2 R I^{-1} \\ -I^{-1} R & I^{-1} \end{pmatrix}, \tag{84}$$

which is symplectic for $\xi^2 = -1$ or orthogonal for $\xi^2 = +1$ because

$$\mathcal{M}^{-1\,T} \Omega \mathcal{M}^{-1} = \Omega. \tag{85}$$

The equations of motion of the $(p+1)$-forms $\mathbf{E}_\Lambda$ and the Bianchi identities of their $(p+2)$-form field strengths $\mathbf{B}^\Lambda = dF^\Lambda$ can be written in a compact way as Bianchi identities for $F^M$

$$d\left(F^M\right) = \begin{pmatrix} \mathbf{B}^\Lambda \\ \mathbf{E}_\Lambda \end{pmatrix} = 0. \tag{86}$$

These equations are invariant under $\mathrm{GL}(2n, \mathbb{R})$ transformations

$$F^{M\,\prime} = S^M{}_N F^N, \quad \text{or} \quad \mathcal{F}' = \mathcal{S}\mathcal{F}, \tag{87}$$

but the twisted self-duality constraint Eq. (82) is only invariant if, at the same time, the scalar matrix $\mathcal{M}$ transforms as

$$\mathcal{M}' = \left(\Omega \mathcal{S} \Omega^{-1}\right) \mathcal{M} \mathcal{S}^{-1}. \tag{88}$$

This implies that the scalar fields must transform as well.

Since the energy-momentum tensor of the $(p+1)$-form potentials can be written in the form

$$-\tfrac{1}{2} I_{\Sigma\Lambda} \left(\iota_c F^\Sigma \wedge \star F^\Lambda + \xi^2 F^\Sigma \wedge \iota_c \star F^\Lambda\right) = -\tfrac{1}{2} \Omega_{MN} \iota_c F^M \wedge F^N, \tag{89}$$

the Einstein equations will be invariant if

$$\mathcal{S}^T \Omega \mathcal{S} = \Omega \quad \Rightarrow \quad \Omega \mathcal{S} \Omega^{-1} = \mathcal{S}^{-1\,T}, \tag{90}$$

*i.e.* if $\mathcal{S} \in \mathrm{Sp}(2n, \mathbb{R})$ when $\xi^2 = -1$ ($p$ even, $d = 4n$) and $\mathcal{S} \in \mathrm{O}(n, n)$ when $\xi^2 = +1$ ($p$ odd, $d = 4n + 2$). This is a well-known generalization of the Gaillard-Zumino result for $p = 0$, presented in Ref. [40].[22]

Then, we conclude that, under symplectic or orthogonal rotations of the potentials, $\mathcal{M}$ must transform as

$$\mathcal{M}' = \mathcal{S}^{-1\,T} \mathcal{M} \mathcal{S}^{-1}, \tag{91}$$

for the twisted self-duality constraint to be respected. These rotations preserve the energy-momentum tensor of the potentials. If we rewrite their kinetic term in the action in the form

$$\sim \mathrm{Tr} \left( d\mathcal{M}^{-1} \wedge \star d\mathcal{M} \right), \tag{92}$$

the invariance of this kinetic term and of the corresponding energy-momentum tensor is manifest. However, this invariance is only apparent since we have not yet described the action of these transformations on the scalar fields, which only transform via field redefinitions or, equivalently, coordinate transformations in the target space, which take the infinitesimal form

$$\delta_\alpha \phi^x = \alpha k^x(\phi), \tag{93}$$

where $\alpha$ is an infinitesimal parameter and $k^x(\phi)$ is a target space vector field. Since these transformations must preserve the kinetic term, they must be Killing vectors of the target space metric $g_{xy}(\phi)$. If $\{k_A{}^x\}$ is the set of these Killing vectors, the possible transformations are

$$\delta_\alpha \phi^x = \alpha^A k_A{}^x(\phi), \tag{94}$$

where now we have as many independent infinitesimal parameters $\alpha^A$ as Killing vectors. These Killing vectors generate an isometry group $G$ which is, in general smaller than $\mathrm{Sp}(2n, \mathbb{R})$ or $\mathrm{O}(n, n)$.

These transformations act on the scalar matrix $\mathcal{M}$ as

$$\delta_\alpha \mathcal{M} = \alpha^A k_A{}^x \partial_x \mathcal{M}, \tag{95}$$

and, according to the previous discussion, they will lead to an invariance of the equations of motion if they are equivalent to the linear transformations Eq. (91). Infinitesimally

$$\mathcal{S} = 1 + \alpha^A T_A, \tag{96a}$$

$$\delta_\alpha \mathcal{M} = -\alpha^A \left( T_A{}^T \mathcal{M} + \mathcal{M} T_A \right), \tag{96b}$$

where the matrices $T_A$ generate a representation of the isometry group $G$ through $\mathrm{Sp}(2n, \mathbb{R})$ or $\mathrm{O}(n, n)$ matrices.

The condition that the matrix $\mathcal{M}$ must satisfy for $G$ to be a symmetry of the equations of motion and Bianchi identities of the $(p + 1)$-form potentials and of the Einstein equations is, therefore,

$$k_A{}^x \partial_x \mathcal{M} + T_A{}^T \mathcal{M} + \mathcal{M} T_A = 0. \tag{97}$$

It only remains to show that these conditions are also sufficient for the transformations to leave invariant the scalar equations of motion. First, we need to rewrite them in a more symmetric form, using the matrix $\mathcal{M}_{MN}$ and the vector of field strengths $F^M$.

---

[22]See Refs. [14, 39, 41] and references therein.

The simplest invariant that we can construct with these elements

$$-\tfrac{1}{4}\mathcal{M}_{MN}F^M \wedge \star F^N \,,\tag{98}$$

vanishes identically when we use the twisted self-duality constraint Eq. (82) and the preservation of $\Omega$ by $\mathcal{M}$ Eq. (85):

$$\begin{aligned}
-\tfrac{1}{4}\mathcal{M}_{MN}F^M \wedge \star F^N &= -\tfrac{1}{4}\xi^2 \mathcal{M}_{MN}\Omega^{NP}\mathcal{M}_{PQ}F^M \wedge F^Q \\
&= -\tfrac{1}{4}\Omega_{MQ}F^M \wedge F^Q \,,
\end{aligned}\tag{99}$$

which vanishes identically because $\Omega_{MN} = \xi^2 \Omega_{NM}$ while $F^M \wedge F^N = -\xi^2 F^N \wedge F^M$.

However,

$$-\tfrac{1}{4}\partial_z \mathcal{M}_{MN}F^M \wedge \star F^N = -\tfrac{1}{4}\xi^2 \partial_z \mathcal{M}_{MN}\Omega^{NP}\mathcal{M}_{PQ}F^M \wedge F^Q \,,\tag{100}$$

does not vanish because, by taking the derivative of Eq. (85), one can easily see that

$$\left(\partial_z \mathcal{M}\Omega^{-1}\mathcal{M}\right)^T = -\xi^2 \partial_z \mathcal{M}\Omega^{-1}\mathcal{M} \,.\tag{101}$$

A straightforward calculation shows that

$$-\tfrac{1}{4}\partial_z \mathcal{M}_{MN}F^M \wedge \star F^N = -\tfrac{1}{2}\partial_z I_{\Sigma\Lambda}F^\Sigma \wedge \star F^\Lambda - \tfrac{1}{2}\partial_z R_{\Sigma\Lambda}F^\Sigma \wedge F^\Lambda \,,\tag{102}$$

and we can rewrite the scalar equation of motion Eq. (79b) in the form

$$\mathbf{E}_z = -g_{zw}\left[d \star d\phi^w + \Gamma_{xy}{}^w d\phi^x \wedge \star d\phi^y\right] - \tfrac{1}{4}\partial_z \mathcal{M}_{MN}F^M \wedge \star F^N \,.\tag{103}$$

Under the infinitesimal transformations

$$\delta_A \phi^x = k_A{}^x \,, \qquad \delta_A F = T_A F \,,\tag{104}$$

where $F \equiv (F^M)$, the scalar equations of motion (103) transform as

$$\begin{aligned}
\delta_A \mathbf{E}_z = \partial_z k_A{}^v \mathbf{E}_v &- 2\nabla_{(z|}k_{A|w)}\left[d \star d\phi^w + \Gamma_{xy}{}^w d\phi^x \wedge \star d\phi^y\right] \\
&- g_{zw}\left(\nabla_x \nabla_y k^w + k^v R_{vxy}{}^w\right)d\phi^x \wedge \star d\phi^y \\
&- \tfrac{1}{4}\partial_z \left(k_A{}^v \partial_v \mathcal{M}_{PQ} + \mathcal{M}_{MQ}T_A{}^M{}_P + \mathcal{M}_{PN}T_A{}^N{}_Q\right)F^P \wedge \star F^Q \,.
\end{aligned}\tag{105}$$

The second and third lines vanish when $k_A{}^x$ is a Killing vector of the target space metric, while the fourth vanishes upon use of the condition Eq. (97).

### 5.1.2 Democratic pseudoaction I: The potentials

Using the results of the previous section it is not difficult to make an educated guess for the democratic pseudoaction that contains the original and dual potentials as independent variables:

$$S[e^a, A^M, \phi^x] = \int \left\{-\star(e^a \wedge e^b) \wedge R_{ab} + \tfrac{1}{2}g_{xy}d\phi^x \wedge \star d\phi^y - \tfrac{1}{4}\mathcal{M}_{MN}F^M \wedge \star F^N\right\} \,.\tag{106}$$

Observe that the last term vanishes automatically when we use the twisted self-duality constraint. However, we are only going to impose it on the equations of motion, which are given by

$$\begin{aligned}
\mathbf{E}_c = \iota_c \star \left(e^a \wedge e^b\right) \wedge R_{ab} &+ \tfrac{1}{2}g_{xy}\left(\iota_c d\phi^x \wedge \star d\phi^y + d\phi^x \wedge \iota_c \star d\phi^y\right) \\
&- \tfrac{1}{4}\mathcal{M}_{MN}\left(\iota_c F^M \wedge \star F^N + \xi^2 F^M \wedge \iota_c \star F^N\right) \,,
\end{aligned}\tag{107a}$$

$$\mathbf{E}_z = -g_{zw}\left[d \star d\phi^w + \Gamma_{xy}{}^w d\phi^x \wedge \star d\phi^y\right] - \tfrac{1}{4}\partial_z \mathcal{M}_{MN}F^M \wedge \star F^N \,,\tag{107b}$$

$$\mathbf{E}_M = \tfrac{1}{2}d\left(\mathcal{M}_{MN}\star F^N\right) \,.\tag{107c}$$

Observe that scalar equation of motion has exactly the same form as the one coming from the original action Eq. (103) and this form will not change when we use the twisted self-duality constraint. Using this constraint in the last equation and using Eq. (85) we see that it takes the form Eq. (86). Finally, using the constraint in the Einstein equations brings the energy-momentum tensor of the potentials to the form Eq. (89). Thus, all the original equations of motion are recovered upon use of the twisted self-duality constraint.

Let us now consider the dualization of the scalars.

### 5.1.3 Democratic pseudoaction II: The scalars

Following the general prescription, we start by computing the Noether-Gaillard-Zumino currents in the democratic theory we have just constructed in which $F^M = dA^M$. Hitting the scalar equations of motion with the Killing vectors $k_A{}^z$ Eq. (107b) and using the Killing vector equations and the condition Eq. (97) we get

$$
\begin{aligned}
k_A{}^z \mathbf{E}_z &= -k_{Aw}\left[d \star d\phi^w + \Gamma_{xy}{}^w d\phi^x \wedge \star d\phi^y\right] - \tfrac{1}{4} k_A{}^z \partial_z \mathcal{M}_{MN} F^M \wedge \star F^N \\
&= -d \star \hat{k}_A + \tfrac{1}{2} T_A{}^M{}_P F^P \wedge \mathcal{M}_{MQ} \star F^Q \\
&= -d\left[\star \hat{k}_A - \tfrac{1}{2} T_A{}^M{}_P A^P \wedge \mathcal{M}_{MQ} \star F^Q\right] + (-1)^p T_A{}^M{}_P A^P \wedge \mathbf{E}_M \,.
\end{aligned}
\tag{108}
$$

Using the twisted self-duality constraint we get a more conventional form

$$
k_A{}^z \mathbf{E}_z = -d\left[\star \hat{k}_A - \tfrac{1}{2} T_A{}^M{}_P \Omega_{MN} A^P \wedge F^N\right] + (-1)^p T_A{}^M{}_P A^P \wedge \mathbf{E}_M \,,
\tag{109}
$$

which leads to on-shell conserved NGZ currents

$$
J_A \equiv \star \hat{k}_A - \tfrac{1}{2} T_A{}^M{}_P \Omega_{MN} A^P \wedge F^N \,,
\tag{110}
$$

and to the definition of the dual $(d-2)$-forms $C_A$

$$
dC_A \equiv \star \hat{k}_A - \tfrac{1}{2} T_A{}^M{}_P \Omega_{MN} A^P \wedge F^N \,,
\tag{111}
$$

and of their gauge-invariant field strengths $G_A$

$$
G_A \equiv dC_A + \tfrac{1}{2} T_A{}^M{}_P \Omega_{MN} A^P \wedge F^N = \star \hat{k}_A \,.
\tag{112}
$$

The gauge invariance of $G_A$ follows from that of the Killing vector and implies that the dual $(d-2)$-form potentials $C_A$ and the $(p+1)$-form potentials $A^M$ transform as

$$
\delta_\Lambda A^M = d\Lambda^M \,, \qquad \delta_\Lambda C_A = d\Lambda_A - \tfrac{1}{2} T_A{}^M{}_P \Omega_{MN} \Lambda^P \wedge F^N \,,
\tag{113}
$$

where $\Lambda^M$ and $\Lambda_A$ are arbitrary $p$- and $(d-3)$-forms, respectively.

Based on our previous results, we propose the fully democratic pseudoaction

$$
\begin{aligned}
S[e^a, A^M, \phi^x, C_A] = \int \Big\{ &-\star(e^a \wedge e^b) \wedge R_{ab} + \tfrac{1}{4} g_{xy} d\phi^x \wedge \star d\phi^y \\
&-\tfrac{1}{4} \mathcal{M}_{MN} F^M \wedge \star F^N + \tfrac{1}{4} \mathfrak{M}^{AB} G_A \wedge \star G_B - \tfrac{1}{2} g^{AB} G_A \wedge \hat{k}_B \Big\} \,,
\end{aligned}
\tag{114}
$$

whose equations of motion have to be supplemented by the twisted self-duality constraint Eq. (82) and by the duality relation Eq. (112).

The equations of motion that follow from the above action are

$$\mathbf{E}_c = \iota_c \star \left( e^a \wedge e^b \right) \wedge R_{ab} + \tfrac{1}{2} g_{xy} \left( \iota_c d\phi^x \wedge \star d\phi^y + d\phi^x \wedge \iota_c \star d\phi^y \right)$$
$$- \tfrac{1}{4} \mathcal{M}_{MN} \left( \iota_c F^M \wedge \star F^N + \xi^2 F^M \wedge \iota_c \star F^N \right)$$
$$+ \frac{1}{4} \mathfrak{M}^{AB} \left( \iota_c G_A \wedge \star G_B + G_A \wedge \iota_c \star G_B \right), \tag{115a}$$

$$\mathbf{E}_z = -\tfrac{1}{2} g_{zw} \left[ d \star d\phi^w + \Gamma_{xy}{}^w d\phi^x \wedge \star d\phi^y \right] - \tfrac{1}{4} \partial_z \mathcal{M}_{MN} F^M \wedge \star F^N$$
$$+ \tfrac{1}{4} \partial_z \mathfrak{M}^{AB} G_A \wedge \star G_B - \tfrac{1}{2} g^{AB} G_A \wedge d\phi^x \partial_z k_{Bx} - \tfrac{1}{2} d \left( g^{AB} G_A k_{Bz} \right), \tag{115b}$$

$$\mathbf{E}_M = \tfrac{1}{2} d \left( \mathcal{M}_{MN} \star F^N \right) + \tfrac{1}{2} \frac{\delta G_A}{A^M} \wedge \left( \mathfrak{M}^{AB} \star G_B - g^{AB} \hat{k}_B \right), \tag{115c}$$

$$\mathbf{E}^A = -\tfrac{1}{2} d \left( \mathfrak{M}^{AB} \star G_B - g^{AB} \hat{k}_B \right). \tag{115d}$$

Using the duality constraints and following the same steps as in previous sections we recover the equations of motion of the original theory.

# 6 A democratic pseudoaction for $\mathcal{N} = 2B, d = 10$ supergravity

The bosonic fields of $\mathcal{N} = 2B, d = 10$ supergravity are the (Einstein-frame) Zehnbein 1-form $e^a$, a $SL(2, \mathbb{R})$ doublet of 2-forms $\mathcal{B}_i$, $i = 1, 2$, 4-form $\mathcal{D}$, which is a $SL(2, \mathbb{R})$ singlet and whose 5-form field strength is self-dual and the complex scalar $\tau$ that parametrizes a $SL(2, \mathbb{R})/SO(2)$ coset. The relation between these fields and those of the effective action of the type IIB superstring (string-frame Zehnbein $e_s^a$, NSNS and RR 2-forms $B, C^{(2)}$, RR 4-form $C^{(4)}$ and dilaton $\varphi$ and RR 0-form $C^{(0)}$) is [14]

$$e^a = e^{-\varphi/4} e_s^a,$$
$$\tau = C^{(0)} + i e^{-\varphi},$$
$$(\mathcal{B}_i) = \begin{pmatrix} C^{(2)} \\ B \end{pmatrix},$$
$$\mathcal{D} = C^{(4)} - \tfrac{1}{2} B \wedge C^{(2)}.\,^{23} \tag{116}$$

The self-duality of the 5-form field strength forbids the existence of a covariant action free of auxiliary fields and one, if one does not want to deal with auxiliary fields, one must necessarily work with equations of motion. The equations of motion of this theory were first found in Ref. [43] in the Einstein frame and using a $SU(1,1)/U(1)$ formulation of the coset space parametrized by the scalar fields. A pseudoaction which had to be supplemented by the self-duality constraint was first constructed in Ref. [34]. A pseudoaction containing the duals of the 1- and 3-form RR field strengths and some higher RR field strengths was constructed in Ref. [31]. While this action was "RR-democratic" it was certainly not "NSNS-democratic". Furthermore, this incomplete democratization breaks the manifest $SL(2, \mathbb{R})$ symmetry of the theory. Our goal in this section is to improve on those results constructing a manifestly $SL(2, \mathbb{R})$-invariant and fully democratic pseudoaction using the results of the previous sections.

---

[23]Observe that the RR 4-form $C^{(4)}$ is not an $SL(2, \mathbb{R})$ and, as a matter of fact, transforms in a complicated way under those transformations. On the other hand, in the rescaling between the string- and Einstein-frame metrics it is very important to take into account the effect of the constant value of the dilaton at infinity, in order to preserve the standard normalization of the metric at infinity [42]. The relation between the string and the so-called *modified Einstein frame* should, then, be $e^a = e^{-(\varphi - \varphi_\infty)/4} e_s^a$. This leads to the occurrence of factors of powers of $e^{\varphi_\infty/4}$ in in different terms of the action that have to be removed by absorbing them in redefinitions of the rest of the fields. We will not study these redefinitions here because they are not relevant to the construction of the democratic pseudoaction.

The transformation of the scalars under $\mathrm{SL}(2,\mathbb{R})$ can be conveniently described through the symmetric $\mathrm{SL}(2,\mathbb{R})$ matrix

$$\left(\mathcal{M}_{ij}\right) \equiv \frac{1}{\mathfrak{Im}\,\tau}\begin{pmatrix} |\tau|^2 & \mathfrak{Re}\,\tau \\[6pt] \mathfrak{Re}\,\tau & 1 \end{pmatrix}, \tag{117}$$

whose inverse is

$$\left(\mathcal{M}^{ij}\right) \equiv \frac{1}{\mathfrak{Im}\,\tau}\begin{pmatrix} 1 & -\mathfrak{Re}\,\tau \\[6pt] -\mathfrak{Re}\,\tau & |\tau|^2 \end{pmatrix}. \tag{118}$$

If we act with the $\mathrm{SL}(2,\mathbb{R})$ transformation matrix

$$\left(\mathcal{S}^{-1\,i}{}_j\right) \equiv \begin{pmatrix} \alpha & \gamma \\ \beta & \delta \end{pmatrix}, \qquad \alpha\delta - \beta\gamma = +1, \tag{119}$$

on objects with indices

$$\mathcal{B}'_i = \mathcal{B}_j \mathcal{S}^{-1\,j}{}_i, \qquad \mathcal{M}'_{ij} = \mathcal{M}_{kl}\mathcal{S}^{-1\,k}{}_i \mathcal{S}^{-1\,l}{}_j, \qquad \mathcal{M}^{ij\,\prime} = \mathcal{S}^i{}_k \mathcal{S}^j{}_l \mathcal{M}^{kl}, \tag{120}$$

then $\tau$ transforms as

$$\tau' = \frac{\alpha\tau + \beta}{\gamma\tau + \delta}. \tag{121}$$

The field strength of the doublet of 2-forms is the doublet of 3-forms

$$\mathcal{H}_i \equiv d\mathcal{B}_i, \tag{122}$$

while the field strength of the 4-form is the $\mathrm{SL}(2,\mathbb{R})$-invariant 5-form

$$\mathcal{F} \equiv d\mathcal{D} - \tfrac{1}{2}\varepsilon^{ij}\mathcal{B}_i \wedge \mathcal{H}_j. \tag{123}$$

The doublet of 3-form field strengths $\mathcal{H}_i$ and the 5-form $\mathcal{F}$ are invariant under the gauge transformations

$$\delta_\Lambda \mathcal{B}_i = d\Lambda_i, \qquad \delta_\Lambda \mathcal{D} = d\Lambda + \tfrac{1}{2}\varepsilon^{ij}\Lambda_i \wedge \mathcal{H}_j, \tag{124}$$

where $\Lambda$ and $\Lambda_i$ are, respectively, a 4-form and a doublet of 1-forms.

This field strength is constrained to be self-dual

$$\mathcal{F} = \star\mathcal{F},\,^{24} \tag{125}$$

and this condition relates the Bianchi identity

$$d\mathcal{F} + \tfrac{1}{2}\varepsilon^{ij}\mathcal{H}_i \wedge \mathcal{H}_j = 0, \tag{126}$$

to the equation of motion [43]

$$d\star\mathcal{F} + \tfrac{1}{2}\varepsilon^{ij}\mathcal{H}_i \wedge \mathcal{H}_j = 0. \tag{127}$$

---

[24]This constraint is, actually, the equation of motion, *sensu stricto*.

Since it also implies that $\mathcal{F} \wedge \star \mathcal{F} = 0$, this constraint makes it impossible to write a covariant action for the theory. Ignoring it, one can write a pseudoaction which leads to the above equation of motion for $\mathcal{D}$ (and to the right equations of motion for the rest of the fields [34]. This pseudoaction can be written in the manifestly SL(2, $\mathbb{R}$)-invariant form

$$
\begin{aligned}
S[e^a, \tau, \mathcal{B}_i, D] = \int \Big\{ &- \star (e^a \wedge e^b) \wedge R_{ab} + \frac{d\tau \wedge \star d\bar{\tau}}{2(\Im m\, \tau)^2} + \tfrac{1}{2} \mathcal{M}^{ij} \mathcal{H}_i \wedge \star \mathcal{H}_j \\
&+ \tfrac{1}{4} \mathcal{F} \wedge \star \mathcal{F} - \tfrac{1}{4} \varepsilon^{ij} \mathcal{D} \wedge \mathcal{H}_i \wedge \mathcal{H}_j \Big\},
\end{aligned}
\tag{128}
$$

and the equations of motion have to be supplemented by the self-duality constraint Eq. (125).

Our goal is to generalize this pseudoaction to include the 8-form duals of the scalar fields (a SL(2, $\mathbb{R}$) triplet) as well as the 6-form duals of the 2-form fields (a dual SL(2, $\mathbb{R}$) doublet). We start by writing all the equations of motion. It is convenient to define the real scalars $\{\phi^x\}$ by $\tau = C^{(0)} + i e^{-\varphi} \equiv \phi^1 + i\phi^2$, so that

$$
\frac{d\tau \wedge \star d\bar{\tau}}{2(\Im m\, \tau)^2} = \tfrac{1}{2} g_{xy} d\phi^x \wedge \star d\phi^y, \qquad (g_{xy}) = \frac{1}{(\phi^2)^2} \begin{pmatrix} 1 & 0 \\ 0 & 1 \end{pmatrix}.
\tag{129}
$$

Then, the equations of motion take the form

$$
\begin{aligned}
\mathbf{E}_a = {}& \iota_a \star (e^c \wedge e^d) \wedge R_{cd} + \tfrac{1}{2} g_{xy} (\iota_a d\phi^x \wedge \star d\phi^y + d\phi^x \wedge \iota_a \star d\phi^y) \\
&+ \tfrac{1}{2} \mathcal{M}^{ij} (\iota_a \mathcal{H}_i \wedge \star \mathcal{H}_j + \mathcal{H}_i \wedge \iota_a \star \mathcal{H}_j) + \tfrac{1}{4} (\iota_a \mathcal{F} \wedge \star \mathcal{F} + \mathcal{F} \wedge \iota_a \star \mathcal{F}),
\end{aligned}
\tag{130a}
$$

$$
\mathbf{E}_x = -g_{xy} [d \star d\phi^y + \Gamma_{zw}{}^y d\phi^z \wedge \star d\phi^w] + \tfrac{1}{2} \partial_x \mathcal{M}^{ij} \mathcal{H}_i \wedge \star \mathcal{H}_j,
\tag{130b}
$$

$$
\mathbf{E}^i = -\big\{ d (\mathcal{M}^{ij} \star \mathcal{H}_j) + \varepsilon^{ij} \mathcal{H}_j \wedge \mathcal{F} \big\} + \tfrac{1}{2} \varepsilon^{ij} \mathcal{H}_j \wedge (\mathcal{F} - \star \mathcal{F}) + \tfrac{1}{2} \varepsilon^{ij} \mathcal{B}_j \wedge \mathbf{E},
\tag{130c}
$$

$$
\mathbf{E} = -\tfrac{1}{2} \big\{ d \star \mathcal{F} + \tfrac{1}{2} \varepsilon^{ij} \mathcal{H}_i \wedge \mathcal{H}_j \big\}.
\tag{130d}
$$

The Einstein equations and the equations of motion of the 2-forms and 4-form simplify when the self-duality constraint is imposed. In particular, since the equation of motion of the 4-form becomes the Bianchi identity, it is automatically solved. The remaining non-trivial equations of motion are

$$
\begin{aligned}
\mathbf{E}_a = {}& \iota_a \star (e^c \wedge e^d) \wedge R_{cd} + \tfrac{1}{2} g_{xy} (\iota_a d\phi^x \wedge \star d\phi^y + d\phi^x \wedge \iota_a \star d\phi^y) \\
&+ \tfrac{1}{2} \mathcal{M}^{ij} (\iota_a \mathcal{H}_i \wedge \star \mathcal{H}_j + \mathcal{H}_i \wedge \iota_a \star \mathcal{H}_j) + \tfrac{1}{2} \iota_a \mathcal{F} \wedge \mathcal{F},
\end{aligned}
\tag{131a}
$$

$$
\mathbf{E}_x = -g_{xy} [d \star d\phi^y + \Gamma_{zw}{}^y d\phi^z \wedge \star d\phi^w] + \tfrac{1}{2} \partial_x \mathcal{M}^{ij} \mathcal{H}_i \wedge \star \mathcal{H}_j,
\tag{131b}
$$

$$
\mathbf{E}^i = -d (\mathcal{M}^{ij} \star \mathcal{H}_j) - \varepsilon^{ij} \mathcal{H}_j \wedge \mathcal{F},
\tag{131c}
$$

together with the self-duality constraint Eq. (125).

## 6.1 Dualization of the 2-forms

In order to construct the democratic pseudoaction, we start by considering the dualization of the doublet of 2-forms $\mathcal{B}_i$ into a doublet of 6-forms that we will denote by $\mathcal{B}^i$.

The equations of motion of the doublet of 2-forms can be written as a total derivative:

$$
\mathbf{E}^i = -d \big[ \mathcal{M}^{ij} \star \mathcal{H}_j + \varepsilon^{ij} \mathcal{B}_j \wedge \big(d\mathcal{D} - \tfrac{1}{6} \varepsilon^{kl} \mathcal{B}_k \wedge \mathcal{H}_l \big) \big],
\tag{132}
$$

and, therefore, they can be locally solved by identifying the expression in square brackets with $d\mathcal{B}^i$. Then,

$$
\mathcal{M}^{ij} \star \mathcal{H}_j = d\mathcal{B}^i - \varepsilon^{ij} \mathcal{B}_j \wedge \big(d\mathcal{D} - \tfrac{1}{6} \varepsilon^{kl} \mathcal{B}_k \wedge \mathcal{H}_l \big) \equiv \mathcal{H}^i,
\tag{133}
$$

where $\mathcal{H}^i$ is the SL$(2,\mathbb{R})$ doublet of 7-form field strengths, invariant under the gauge transformations Eqs. (124) if the 6-forms transform according to

$$\delta_\Lambda \mathcal{B}^i = d\Lambda^i + \varepsilon^{ij}\Lambda_j \wedge d\mathcal{D} - \tfrac{1}{6}\varepsilon^{ij}\varepsilon^{lk}\mathcal{B}_j \wedge \mathcal{B}_l \wedge d\Lambda_k\,, \tag{134}$$

where $\Lambda^i$ is a doublet of 5-forms.

The Bianchi identity of the 3-form field strengths

$$d\mathcal{H}_i = 0\,, \tag{135}$$

becomes the equation of motion of the 6-forms upon use of the duality relation

$$\mathcal{H}_i = \mathcal{M}_{ij} \star \mathcal{H}^j\,, \tag{136}$$

that is

$$d\left(\mathcal{M}_{ij} \star \mathcal{H}^j\right) = 0\,. \tag{137}$$

The dual 6-forms that we have just defined can easily be included in a semi-democratic pseudoaction,

$$\begin{aligned}
S_{\text{SemiDem}}[e^a,\tau,\mathcal{B}_i,D,\mathcal{B}^i] = \int \Big\{ &-\star(e^a \wedge e^b) \wedge R_{ab} + \tfrac{1}{2}g_{xy}d\phi^x \wedge \star d\phi^y \\
&+ \tfrac{1}{4}\mathcal{M}^{ij}\mathcal{H}_i \wedge \star\mathcal{H}_j + \tfrac{1}{4}\mathcal{F} \wedge \star\mathcal{F} \\
&+ \tfrac{1}{4}\mathcal{M}_{ij}\mathcal{H}^i \wedge \star\mathcal{H}^j + \tfrac{1}{4}\varepsilon^{ij}\mathcal{D} \wedge \mathcal{H}_i \wedge \mathcal{H}_j \Big\}\,.^{[25]}
\end{aligned} \tag{138}$$

The equations of motion that follow from this pseudoaction are

$$\begin{aligned}
\mathbf{E}_a = \;&\iota_a \star (e^c \wedge e^d) \wedge R_{cd} + \tfrac{1}{2}g_{xy}\left(\iota_a d\phi^x \wedge \star d\phi^y + d\phi^x \wedge \iota_a \star d\phi^y\right) \\
&+ \tfrac{1}{4}\mathcal{M}^{ij}\left(\iota_a \mathcal{H}_i \wedge \star\mathcal{H}_j + \mathcal{H}_i \wedge \iota_a \star \mathcal{H}_j\right) + \tfrac{1}{4}\left(\iota_a\mathcal{F} \wedge \star\mathcal{F} + \mathcal{F} \wedge \iota_a \star \mathcal{F}\right) \\
&+ \tfrac{1}{4}\mathcal{M}_{ij}\left(\iota_a \mathcal{H}^i \wedge \star\mathcal{H}^j + \mathcal{H}^i \wedge \iota_a \star \mathcal{H}^j\right)\,,
\end{aligned} \tag{139a}$$

$$\mathbf{E}_x = -g_{xy}\left[d \star d\phi^y + \Gamma_{zw}{}^y d\phi^z \wedge \star d\phi^w\right] + \tfrac{1}{4}\partial_x\mathcal{M}^{ij}\mathcal{H}_i \wedge \star\mathcal{H}_j + \tfrac{1}{4}\partial_x\mathcal{M}_{ij}\mathcal{H}^i \wedge \star\mathcal{H}^j\,, \tag{139b}$$

$$\begin{aligned}
\mathbf{E}^i = \;&-\tfrac{1}{2}\left\{d\left(\mathcal{M}^{ij} \star \mathcal{H}_j\right) + \varepsilon^{ij}\mathcal{H}_j \wedge \mathcal{F}\right\} + \tfrac{1}{2}\varepsilon^{ij}\mathcal{H}_j \wedge (\star\mathcal{F} - \mathcal{F}) \\
&+ \tfrac{1}{2}\varepsilon^{ij}\left(\mathcal{H}_j - \mathcal{M}_{jk} \star \mathcal{H}^k\right) \wedge \mathcal{F} + \tfrac{1}{2}\varepsilon^{ij}\mathcal{B}_j \wedge \mathbf{E} - \tfrac{2}{3}\varepsilon^{ij}\varepsilon^{kl}\mathcal{B}_j \wedge \mathcal{B}_k \wedge \mathbf{E}_l\,,
\end{aligned} \tag{139c}$$

$$\mathbf{E} = -\tfrac{1}{2}\left\{d \star \mathcal{F} - \tfrac{1}{2}\varepsilon^{ij}\mathcal{H}_i \wedge \left(\mathcal{H}_j - 2\mathcal{M}_{jk} \star \mathcal{H}^k\right)\right\} + \varepsilon^{ij}\mathcal{B}_i \wedge \mathbf{E}_j\,, \tag{139d}$$

$$\mathbf{E}_i = -\tfrac{1}{2}d\left(\mathcal{M}_{ij} \star \mathcal{H}^j\right)\,. \tag{139e}$$

Upon use of the duality relations Eq. (136) the equations of motion of the 6-forms $\mathbf{E}_i$ become the Bianchi identities of the 3-forms and are automatically solved. Furthermore, using the same duality relations plus the self-duality constraint Eq. (125) the last two lines and the third term of the first line of the equations of motion of the 2-forms $\mathbf{E}^i$ vanish and what remains becomes, up to a factor of $1/2$, the original equations of motion of the 2-forms. The rest of the equations of motion are trivially recovered.

## 6.2 Dualization of the scalars

The next step towards the construction of the democratic pseudoaction is the dualization of the scalars, which in this theory parametrize the symmetric Riemannian manifold SL$(2,\mathbb{R})$/SO$(2)$. This means that we can use the general procedure described in Sections 4 and 5.

---

[25]Observe that the sign of the Chern-Simons term has been reversed.

Under the three independent infinitesimal $SL(2,\mathbb{R})$ transformations labeled by $A$, the fields that we have introduced so far transform according to

$$\delta_A \phi^x = k_A{}^x(\phi), \qquad \delta_A \mathcal{B}_i = -\mathcal{B}_j (T_A)^j{}_i, \qquad \delta_A \mathcal{B}^i = (T_A)^i{}_j \mathcal{B}_j, \tag{140}$$

while the kinetic matrix $\mathcal{M}_{ij}$ and its inverse $\mathcal{M}^{ij}$ satisfy

$$k_A{}^x \partial_x \mathcal{M}_{ij} = -2\mathcal{M}_{k(j}(T_A)^k{}_{i)}, \qquad k_A{}^x \partial_x \mathcal{M}^{ij} = 2(T_A)^{(i}{}_k \mathcal{M}^{j)k}. \tag{141}$$

Observe that due to the fact that $\varepsilon^{ij}$ and $\varepsilon_{ij}$ are $SL(2,\mathbb{R})$-invariant tensors, the matrices $T_A$ satisfy

$$(T_A)^{[i}{}_k \varepsilon^{j]k} = (T_A)^k{}_{[i} \varepsilon_{j]k} = 0. \tag{142}$$

Then, we can obtain the Noether-Gaillard-Zumino 9-form currents $J_A$ using the Killing vector equation, the duality relations and the equations of motion of the 2- and 6-forms and these properties, obtaining

$$
\begin{aligned}
k_A{}^x \mathbf{E}_x = -d\Big[ &\star \hat{k}_A - \tfrac{1}{2}(T_A)^i{}_k \mathcal{B}_i \wedge \mathcal{H}^k + \tfrac{1}{2}(T_A)^k{}_i \mathcal{B}^i \wedge \mathcal{H}_k \\
&+ \tfrac{1}{24}(T_A)^i{}_k \varepsilon^{kj} \varepsilon^{mn} \mathcal{B}_i \wedge \mathcal{B}_j \wedge \mathcal{B}_m \wedge \mathcal{H}_n \Big].
\end{aligned}
\tag{143}
$$

This expression vanishes on-shell, and we can solve it locally by introducing a $SL(2,\mathbb{R})$ triplet of 8-forms $C_A$ whose exterior derivative equals the expression in brackets. Since the Killing vectors are gauge invariant, we can define the following gauge-invariant triplet of 9-form fields strengths

$$
\begin{aligned}
\star \hat{k}_A = {} & dC_A + \tfrac{1}{2}(T_A)^i{}_k \mathcal{B}_i \wedge \mathcal{H}^k - \tfrac{1}{2}(T_A)^k{}_i \mathcal{B}^i \wedge \mathcal{H}_k \\
& - \tfrac{1}{24}(T_A)^i{}_k \varepsilon^{kj} \varepsilon^{mn} \mathcal{B}_i \wedge \mathcal{B}_j \wedge \mathcal{B}_m \wedge \mathcal{H}_n \\
\equiv {} & G_A.
\end{aligned}
\tag{144}
$$

$G_A$ is invariant under the gauge transformations in Eqs. (124) and (134) if the 8-forms transform as

$$\delta C_A = d\Lambda_A - \tfrac{1}{2}(T_A)^i{}_k \left\{ \Lambda^k \wedge \mathcal{H}_i - \Lambda_i \wedge \mathcal{H}^k + \tfrac{1}{4}\varepsilon^{kl}\varepsilon^{mn} \mathcal{B}_i \wedge \mathcal{B}_l \wedge \Lambda_m \wedge \mathcal{H}_n \right\}. \tag{145}$$

Having defined the 8-form duals of the scalars and using our previous experience, we propose the following manifestly gauge- and $SL(2,\mathbb{R})$ fully democratic pseudoaction

$$
\begin{aligned}
S_{\text{Dem}}[e^a, \tau, \mathcal{B}_i, D, \mathcal{B}^i, C_A] = \int \Big\{ & -\star(e^a \wedge e^b) \wedge R_{ab} + \tfrac{1}{4}g_{xy} d\phi^x \wedge \star d\phi^y + \tfrac{1}{4}\mathcal{M}^{ij}\mathcal{H}_i \wedge \star \mathcal{H}_j \\
& + \tfrac{1}{4}\mathcal{F} \wedge \star \mathcal{F} + \tfrac{1}{4}\mathcal{M}_{ij}\mathcal{H}^i \wedge \star \mathcal{H}^j + \tfrac{1}{4}\mathfrak{M}^{AB} G_A \wedge \star G_B \\
& - \tfrac{1}{2}g^{AB} G_A \wedge \star \hat{k}_B + \tfrac{1}{4}\varepsilon^{ij} \mathcal{D} \wedge \mathcal{H}_i \wedge \mathcal{H}_j \Big\}.
\end{aligned}
\tag{146}
$$

The explicit form of this action depends on the particular choice of Killing vector basis. A convenient choice is

$$k_1 = C^{(0)}\partial_C - \partial_\varphi, \qquad k_2 = (e^{-2\varphi} - C^{(0)2})\partial_C + 2C^{(0)}\partial_\varphi, \qquad k_3 = \partial_C. \tag{147}$$

Since, in this basis, $k_3$ generates the constant shifts of the RR 0-form $C^{(0)}$, we can identify the 8-form $C_3$ with the RR 8-form $C^{(8)}$.

The Lie brackets of these vectors are

$$[k_1, k_3] = -k_3, \qquad [k_2, k_3] = 2k_1, \qquad [k_1, k_2] = k_2, \tag{148}$$

which leads to the Killing metric $K_{AB}$

$$K_{AB} = f_{AC}{}^D f_{BD}{}^C = \begin{pmatrix} 2 & 0 & 0 \\ 0 & 0 & 4 \\ 0 & 4 & 0 \end{pmatrix}. \tag{149}$$

In our conventions the metric $g_{AB}$ used to construct the $\sigma$-model metric is then related with the Killing metric by

$$g_{AB} = \tfrac{1}{2}K_{AB} = \begin{pmatrix} 1 & 0 & 0 \\ 0 & 0 & 2 \\ 0 & 2 & 0 \end{pmatrix}. \tag{150}$$

Then the matrix $\mathfrak{M}^{AB}$ defined in Eq. (44) is given by

$$\left(\mathfrak{M}^{AB}\right) = \begin{pmatrix} 1 + e^{2\varphi}C^{(0)2} & \tfrac{1}{2}e^{2\varphi}C^{(0)} & -\tfrac{1}{2}C^{(0)}\left[1 + e^{2\varphi}C^{(0)2}\right] \\ \tfrac{1}{2}e^{2\varphi}C^{(0)} & \tfrac{1}{4}e^{2\varphi} & \tfrac{1}{4}\left[1 - e^{2\varphi}C^{(0)2}\right] \\ -\tfrac{1}{2}C^{(0)}\left[1 + e^{2\varphi}C^{(0)2}\right] & \tfrac{1}{4}\left[1 - e^{2\varphi}C^{(0)2}\right] & \tfrac{1}{4}e^{-2\varphi}\left[1 + e^{2\varphi}C^{(0)2}\right]^2 \end{pmatrix}. \tag{151}$$

We have explicitly checked that it satisfies the essential property Eq. (46).

The equations of motion that follow from the democratic pseudoaction Eq. (146) are

$$
\begin{aligned}
\mathbf{E}_a &= \iota_a \star (e^c \wedge e^d) \wedge R_{cd} + \tfrac{1}{4}g_{xy}\left(\iota_a d\phi^x \wedge \star d\phi^y + d\phi^x \wedge \iota_a \star d\phi^y\right) \\
&\quad + \tfrac{1}{4}\mathcal{M}^{ij}\left(\iota_a \mathcal{H}_i \wedge \star \mathcal{H}_j + \mathcal{H}_i \wedge \iota_a \star \mathcal{H}_j\right) + \tfrac{1}{4}\left(\iota_a \mathcal{F} \wedge \star \mathcal{F} + \mathcal{F} \wedge \iota_a \star \mathcal{F}\right) \\
&\quad + \tfrac{1}{4}\mathcal{M}_{ij}\left(\iota_a \mathcal{H}^i \wedge \star \mathcal{H}^j + \mathcal{H}^i \wedge \iota_a \star \mathcal{H}^j\right) + \tfrac{1}{4}\mathfrak{M}^{AB}\left(\iota_a G_A \wedge \star G_B + G_A \iota_a \star G_B\right),
\end{aligned} \tag{152a}
$$

$$
\begin{aligned}
\mathbf{E}_x &= -g_{xy}\left[d \star d\phi^y + \Gamma_{zw}{}^y d\phi^z \wedge \star d\phi^w\right] + \tfrac{1}{4}\partial_x \mathcal{M}^{ij}\mathcal{H}_i \wedge \star \mathcal{H}_j + \tfrac{1}{4}\partial_x \mathcal{M}_{ij}\mathcal{H}^i \wedge \star \mathcal{H}^j \\
&\quad + \tfrac{1}{2}g^{AC}\partial_x k_{C\,y}G_A \wedge \left[g^{BD}k_D{}^y \star G_B - d\phi^y\right],
\end{aligned} \tag{152b}
$$

$$
\begin{aligned}
\mathbf{E}^i &= -\tfrac{1}{2}\left\{d\left(\mathcal{M}^{ij} \star \mathcal{H}_j\right) + \varepsilon^{ij}\mathcal{H}_j \wedge \mathcal{F}\right\} + \tfrac{1}{2}\varepsilon^{ij}\mathcal{H}_j \wedge (\star \mathcal{F} - \mathcal{F}) \\
&\quad + \tfrac{1}{2}\varepsilon^{ij}\left(\mathcal{H}_j - \mathcal{M}_{jk} \star \mathcal{H}^k\right) \wedge \mathcal{F} + \tfrac{1}{2}\frac{\delta G_A}{\delta \mathcal{B}_i} \wedge \left[\mathfrak{M}^{AB} \star G_B - \hat{k}_B\right] \\
&\quad + \tfrac{1}{2}\varepsilon^{ij}\mathcal{B}_j \wedge \mathbf{E} - \tfrac{2}{3}\varepsilon^{ij}\varepsilon^{kl}\mathcal{B}_j \wedge \mathcal{B}_k \wedge \mathbf{E}_l,
\end{aligned} \tag{152c}
$$

$$
\begin{aligned}
\mathbf{E} &= -\tfrac{1}{2}\left\{d \star \mathcal{F} - \tfrac{1}{2}\varepsilon^{ij}\mathcal{H}_i \wedge \left(\mathcal{H}_j - 2\mathcal{M}_{jk} \star \mathcal{H}^k\right)\right\} + \tfrac{1}{2}\frac{\delta G_A}{\delta \mathcal{D}} \wedge \left[\mathfrak{M}^{AB} \star G_B - \hat{k}_B\right] \\
&\quad + \varepsilon^{ij}\mathcal{B}_i \wedge \mathbf{E}_j,
\end{aligned} \tag{152d}
$$

$$\mathbf{E}_i = -\tfrac{1}{2}d\left(\mathcal{M}_{ij} \star \mathcal{H}^j\right) + \tfrac{1}{2}\frac{\delta G_A}{\delta \mathcal{B}^i} \wedge \left[\mathfrak{M}^{AB} \star G_B - \hat{k}_B\right], \tag{152e}$$

$$\mathbf{E}^A = -\tfrac{1}{2}d\left[\mathfrak{M}^{AB} \star G_B - \hat{k}_B\right]. \tag{152f}$$

Using the duality relations and the results obtained in the previous sections we recover the original equations of motion of $\mathcal{N} = 2B, d = 10$ supergravity.

# 7 Conclusions

The results obtained in this paper and, in particular, the democratic and manifestly duality-invariant pseudoactions of $d = 4$ maximal and half-maximal supergravities and of $\mathcal{N} = 2B, d = 10$ supergravity can be used in different ways. For instance

1. They can be used to revisit many of the results on flux compactifications and gauged supergravities in a duality-invariant form [44, 45].

2. They can be used to study black-hole thermodynamics using Euclidean methods [46].[26]

3. They can be used to improve our understanding of the interplay between T and S dualities. In particular, one can improve our understanding of the duality between type IIB 7- and type IIA 8-branes [15, 45, 48].

However, several extensions of the results presented here are still necessary:

1. The supersymmetry transformation rules of all the dual fields we have introduced should be found.

2. The 10-forms which are known to exist and play an important role in $\mathcal{N} = 2B, d = 10$ supergravity [49] should be added somehow to the pseudoaction in order to have a complete picture of the dualities between fields and fluxes.

Work on some of these directions is in progress.

## Acknowledgments

TO wishes to thank M.M. Fernández for her permanent support.

**Funding information**    The work of JJF-M and GG has been supported in part by the MCI, AEI, FEDER (UE) grant PID2021-125700NAC22. The work of CG-F, TO and MZ has been supported in part by the MCI, AEI, FEDER (UE) grants PID2021-125700NB-C21 ("Gravity, Supergravity and Superstrings" (GRASS)) and IFT Centro de Excelencia Severo Ochoa CEX2020-001007-S. The work of GG has been supported by the predoctoral fellowship FPI-UM R-1006-2021-01. The work of CG-F has been supported by the MU grant FPU21/02222. The work of MZ has been supported by the fellowship LCF/BQ/DI20/11780035 from "La Caixa" Foundation (ID 100010434).

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
