# Peer review of "Democratic actions with scalar fields: symmetric sigma models, supergravity actions and the effective theory of the type IIB superstring"

_SciPost Physics Core, doi:SciPost Phys. Core 7, 068 (2024)_

## Round 2 · Referee Report · Anonymous (Referee 1) · 2024-7-16

Strengths
1) The paper systematically addresses different dualisations in Supergravity theories. 2) The authors suggest symmetry-enhancing formulations using different pseudoactions.
Weaknesses
1) The paper concentrates only on the pseudo-action approaches, while several approaches are available currently for democratic formulations with proper actions (using auxiliary fields). 2) Some comments about the earlier literature are not completely correct.
Report
We have found a few minor misprints or incorrect statements.
Requested changes
1) In footnote 3, the authors refer to references [18,19] as related to non-linear electrodynamics. The reference [19], however, covers democratic Lagrangians for arbitrary p-forms (and their duals) in arbitrary dimensions. The results of [19] could be used to formulate proper actions for the systems described in the current work via pseudoactions. For the free fields, this approach was introduced in K. Mkrtchyan '19 and discussed in detail for arbitrary p-forms and (d-p-2)-form duals in S. Bansal, O. Evnin, K. Mkrtchyan '21.
2) In the third paragraph of page 3, "... relations between (d + 1)- and (d − p − 3)-forms..." should be "... relations between (p + 1)- and (d − p − 3)-forms...".
3) Statements like "The self-duality of the 5-form field strength forbids the existence of a covariant action and one must necessarily work with equations of motion." (page 28) are incorrect, given that there are covariant actions for self-dual fields, some of which are quoted in this work. We believe the authors want to stress the absence of covariant actions without auxiliary fields, but the current statement is too strong.
4) In the next to last paragraph of page 36, "... known to exist an play an important role..." should be "... known to exist and play an important role..."
Recommendation
Ask for minor revision

Author: Tomás Ortín on 2024-08-08 [id 4680]
(in reply to Report 1 on 2024-07-16)We would like to thank the referee for the constructive comments.
We have implemented all the referee's suggestions, adding the references mentioned in his/her first comment, modifying the sentence mentioned in the third comment and fixing the misprints mentioned in the other two comments.
We have also added another reference that reviews the PST method:
C. Ferko, S. M. Kuzenko, K. Lechner, D. P. Sorokin and G. Tartaglino-
Mazzucchelli, “Interacting chiral form field theories and TT-like flows in six
and higher dimensions,” JHEP 05 (2024), 320 [arXiv:2402.06947 [hep-th]].
This reference complements reference [23] in the previous version (which now has become [25] after the addition of the other two references. These references are in footnote 4 in page 3.
Finally, we have corrected four equations: 5.24c, 5.24d, 5.37c and 5.37d which contained some errors that did not influence the main results of the paper.
We hope the paper can be now be published.
Attachment:
DUALSCALARS.pdf

---

## Round 2 · Referee Report · Anonymous (Referee 2) · 2024-7-18

Strengths
Report
As authors notice in the Introduction, the 'true' actions with auxiliary fields have been known for such cases, including IIB supergravity [21,22], in the frame of PST approach [20] and other approaches some of which were reviewed in [23] (and more in recent [JHEP05(2024)320 e-Print: 2402.06947 [hep-th]]). However, the pseudoaction approach of this manuscript, which does not involve auxiliary fields but imply imposing of generalized duality and self-duality conditions by hand, after obtaining equations of motion [28], may be useful in some circumstances, particularly due to its relative simplicity, and hence also deserves to be developed.
The presentation is very good and clear. I recommend this paper for publication in Science Post in its present form.
Recommendation
Publish (easily meets expectations and criteria for this Journal; among top 50%)

---

## Editorial Decision

published